# Modeled grid cells aligned by a flexible attractor

**Sabrina Benas[1†], Ximena Fernandez[2†], Emilio Kropff[1]\***

[1]Leloir Institute – IIBBA/CONICET, Buenos Aires, Argentina; [2]Department of Mathematics, Durham University, Durham, United Kingdom

## eLife assessment

In this **valuable** study, the authors use a computational model to investigate how recurrent connections influence the firing patterns of grid cells, which are thought to play a role in encoding an animal's position in space. The work suggests that a one-dimensional network architecture may be sufficient to generate the hexagonal firing patterns of grid cells, a possible alternative to attractor models based on recurrent connectivity between grid cells. However, the support for this proposal was **incomplete**, as some conclusions for how well the model dynamics are necessary to generate features of grid cell organization were not well supported.

**\*For correspondence:**
kropff@gmail.com

[†]These authors contributed equally to this work

**Abstract** Entorhinal grid cells implement a spatial code with hexagonal periodicity, signaling the position of the animal within an environment. Grid maps of cells belonging to the same module share spacing and orientation, only differing in relative two-dimensional spatial phase, which could result from being part of a two-dimensional attractor guided by path integration. However, this architecture has the drawbacks of being complex to construct and rigid, path integration allowing for no deviations from the hexagonal pattern such as the ones observed under a variety of experimental manipulations. Here, we show that a simpler one-dimensional attractor is enough to align grid cells equally well. Using topological data analysis, we show that the resulting population activity is a sample of a torus, while the ensemble of maps preserves features of the network architecture. The flexibility of this low dimensional attractor allows it to negotiate the geometry of the representation manifold with the feedforward inputs, rather than imposing it. More generally, our results represent a proof of principle against the intuition that the architecture and the representation manifold of an attractor are topological objects of the same dimensionality, with implications to the study of attractor networks across the brain.

## Introduction

Grid cells in the medial entorhinal cortex and other brain areas provide a representation of the spatial environment navigated by an animal, through maps of hexagonal periodicity that have been compared to a system of Cartesian axes (*Moser et al., 2008*; *Fyhn et al., 2004*; *Buzsáki and Moser, 2013*). While different mechanisms have been proposed as the basis to make a neuron develop a collection of responsive fields distributed in space with hexagonal periodicity, the alignment of axes of symmetry between neighboring, co-modular neurons in most computational models occurs through local synaptic interactions between them (*Kropff and Treves, 2008*; *Couey et al., 2013*; *Burak and Fiete, 2009*; *Burgess et al., 2007*). In general, the network responsible for this communication can be thought of as a two-dimensional continuous attractor (*Knierim and Zhang, 2012*). Models tend to focus either on grid cells performing path integration (*Barry et al., 2007*; *Couey et al., 2013*; *Burak and Fiete, 2009*; *Burgess et al., 2007*) or, as in the case of this work, on mapping of spatial inputs.

Attractor networks are among the clearest examples of unsupervised self-organization in the brain. Point-like attractors emerge naturally in a network with dense recurrent connectivity equipped with Hebbian plasticity, and can be used to store and retrieve discrete pieces of information (*Hopfield, 1982*). If a number of point-like attractors is set close enough to each other along some manifold, a continuous attractor emerges. One-dimensional ring attractors have been used to model head direction cells (*Redish et al., 1996*; *Zhang, 1996*), while two dimensional attractors have been used to model population maps of space such as those of place cells or grid cells (*Burak and Fiete, 2009*; *Battaglia and Treves, 1998*; *Gardner et al., 2022*). A common underlying assumption is that the dimensionality of the network architecture mimics that of the space that is being represented, which explains why the word 'dimensionality' applied to an attractor is indistinctly used to refer to one or the other. However, the network activity does not only depend on recurrent connections, but also on inputs, and the potential interplay between these two sources has so far received little attention. Grid cells have been modeled using two-dimensional attractors because they represent two-dimensional space, but a number of reasons call for the exploration of alternatives. First, grid cells are also capable of representing one dimensional variables such as space, time or the frequency of a sound, or three-dimensional space, exhibiting poor to no periodicity (*Aronov et al., 2017*; *Kraus et al., 2015*; *Yoon et al., 2016*; *Hafting et al., 2008*; *Grieves et al., 2021*; *Ginosar et al., 2021*). Second, two-dimensional attractors impose a rather rigid constraint on the activity of neurons, but grid maps can suffer global or local modifications in response to different experimental manipulations (*Barry et al., 2007*; *Yoon et al., 2013*; *Krupic et al., 2015*; *Krupic et al., 2018*; *Boccara et al., 2019*; *Butler et al., 2019*; *Sanguinetti-Scheck and Brecht, 2020*). While distortions do not necessarily speak against attractor activity, they are difficult to explain from the point of view of attractors purely guided by path integration. Third, the mechanisms behind the formation and maintenance of such a complex and fine-tuned network are far from understood, and theoretical proposals tend to involve as a prerequisite an independent functional representation of two-dimensional space to serve as a tutor (*Si and Treves, 2013*; *Widloski and Fiete, 2014*). Fourth, a recent experiment shows that when animals are trained to navigate deprived from sensory and vestibular feedback, entorhinal cells tend to develop a surprising ring (1D) rather than toroidal (2D) population dynamic (*Gonzalo Cogno et al., 2024*).

Here, we explore the possibility that grid cells are aligned by simpler, one-dimensional attractors, which, as we show, have the potential to flexibly organize the population activity into a space with a dimensionality that is negotiated with the inputs rather than pre-defined. Crucially, we show for the

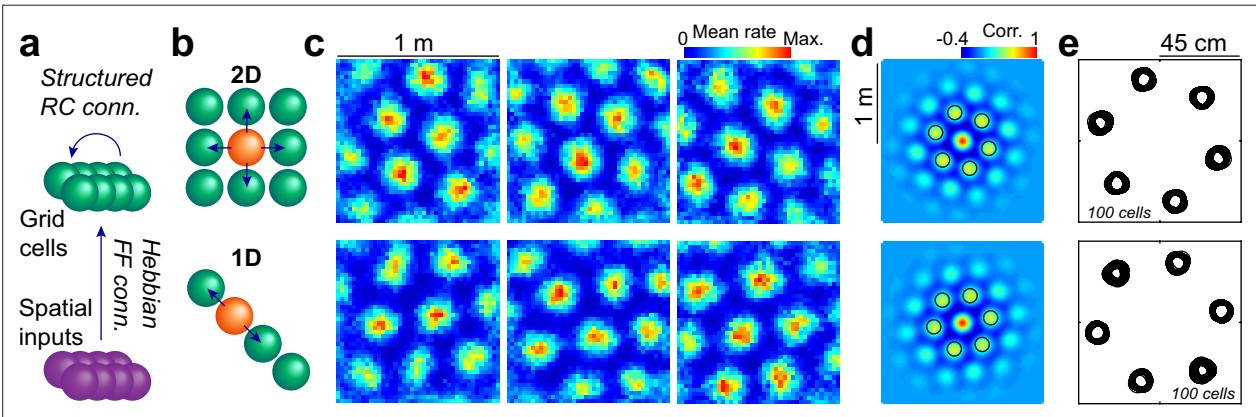

**Figure 1.** Attractors with a 2D or 1D architecture align grid maps. (**a**) Schematics of the network model, including an input layer with place cell-like activity (purple), feedforward all-to-all connections with Hebbian plasticity and a grid cell layer (green) with global inhibition and a set of excitatory recurrent collaterals of fixed structure. (**b**) Schematics of the recurrent connectivity from a given cell (orange) to its neighbors (green) in a 2D (top) or 1D (bottom) setup. (**c**) Representative examples of maps belonging to the same 2D (top) or 1D (bottom) network at the end of training. (**d**) Average of all autocorrelograms in the same two simulations, highlighting the 6 maxima around the center (black circles). (**e**) Superposition of the 6 maxima around the center (as in **d**) for all individual autocorrelograms in the same two simulations.

The online version of this article includes the following figure supplement(s) for figure 1:

**Figure supplement 1.** Spatial phases present characteristic patterns for simple attractors, although this is not a necessary feature for slightly more complex architectures.

first time with mathematical rigor that the architecture and representational space of an attractor network can be two different topological objects. This proof of principle broadens the spectrum of potential candidates for the recurrent architecture interconnecting grid cells, opening the possibility of variability along animal development and maturation, or across the multiple brain areas where grid cells have been described.

## Results

### Grid maps aligned by a one-dimensional attractor

To understand if grid maps can be aligned by an architecture of excitatory recurrent collateral connections simpler than a two-dimensional attractor, we trained a model in which grid maps are obtained from spatial inputs through self-organization (*Kropff and Treves, 2008*). In this model, a layer of spatially stable inputs projects to a layer of grid cells through feedforward connections equipped with Hebbian plasticity (*Figure 1a*). Two factors, all-to-all inhibition and adaptation, force neurons in the grid cell layer to take turns to get activated. This dynamic promotes selectivity in the potentiation of afferent synapses to any given grid cell. As a virtual animal navigates an open-field environment, modeled entorhinal cells self-organize, acquiring maps with hexagonal symmetry as a result of Hebbian sculpting of the feedforward connectivity matrix. Previous work shows that these maps are not naturally aligned unless excitatory recurrent collaterals are included (*Kropff and Treves, 2008*; *Si and Treves, 2013*).

We performed 100 simulations of a simplified version of this self-organizing network (Methods), including 225 input cells and $N_{EC} = 100$ grid cells, in two otherwise identical setups. In the first scenario (2D), we added to the grid cell layer a classical architecture of recurrent collateral connections shaped as a torus (*Figure 1b*). In the second scenario (1D), we used instead a much simpler ring attractor architecture. At the end of the learning process, maps in both types of simulation had hexagonal

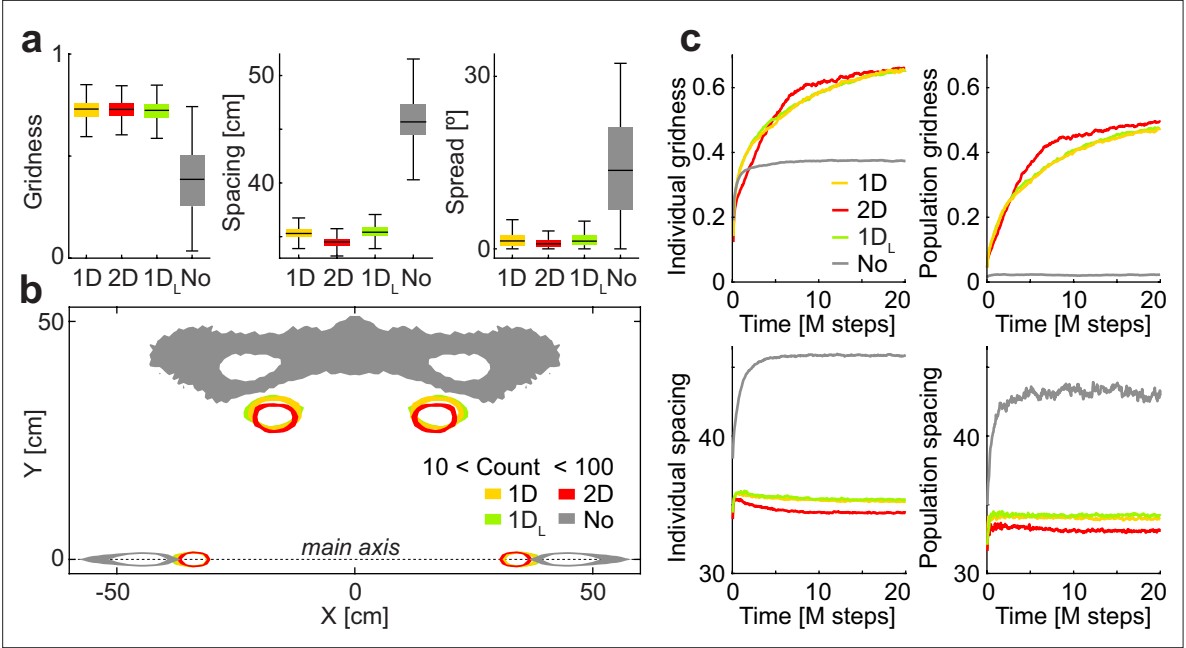

**Figure 2.** Quantification of the alignment and contraction of grid maps by different attractor architectures. (**a**) Distribution (i.q.r., n = 10000) of gridness (left), spacing (center), and spread (right) at the end of the learning process across conditions (quartiles; identical simulations except for the architecture of recurrent collaterals). (**b**) Smoothed distribution of maxima relative to the main axis of the corresponding autocorrelogram. (**c**) Mean evolution of gridness (top) and spacing (bottom) in transient maps along the learning process, calculated from individual (left) or average (right) autocorrelograms. Individual spacing negatively correlates with time for the 2D (R: –0.78, p: 10⁻⁸⁴), 1D (R: –0.77, p: 10⁻⁷⁹), and 1 DL (R: –0.74, p: 10⁻⁷¹).

The online version of this article includes the following figure supplement(s) for figure 2:

**Figure supplement 1.** Too weak, too strong or shuffled attractors fail to align grid cells.

**Figure supplement 2.** Instant improvement in gridness by turning off recurrent collaterals is reverted by learning.

symmetry (*Figure 1c*). The mean population autocorrelogram also had hexagonal symmetry, indicating that individual maps within the network shared spacing and orientation (*Figure 1d*), which was further confirmed by the clustering into six well-defined groups of first-order autocorrelogram maxima for the pool of all cells (*Figure 1e*), with phases were distributed in distinctive patterns for both conditions (*Figure 1—figure supplement 1*). Similar alignment was obtained for the 2D architecture, which was constructed ad hoc for this purpose, and for the much simpler 1D architecture.

For a quantitative comparison of grid cell properties, we incorporated two additional conditions: a stripe-like linear attractor ($1_{DL}$), similar to 1D but with no periodic boundaries, and a condition with no recurrent collaterals (No), in setups otherwise identical to those of 1D and 2D. We compared across conditions the hexagonality of maps (through a gridness index), the spacing between neighboring fields and the angular spread in axes of symmetry across the population, indicative of alignement (*Figure 2a*). We found marked differences only between the No condition and the other three. Gridness was highest for 1D, followed closely by 2D and $1_{DL}$, while the No condition exhibited markedly lower values. Spacing and spread were lowest for the 2D condition, followed by a small margin by 1D and $1_{DL}$, with the No condition again presenting the largest differences with the rest. These results suggest that attractor connectivity of all investigated types not only aligns grid cells similarly but also has the effect of compressing maps in terms of spacing. To visualize how individual maps varied across categories, we plotted the distribution of pooled maxima (as in *Figure 1e*) relative to the axis of the corresponding autocorrelogram presenting the highest correlation value (*Figure 2b*).

To address differences in the self-organization dynamics, we next inspected maps along the learning process (*Figure 2c*). We found that the mean gridness of cells initially increased at a similar pace for all conditions, saturating early in this process only for the No condition. Further increase in gridness was an emergent property only allowed by attractor dynamics, which in the 2D condition took a slightly slower start compensated later by an elbow toward plateauing behavior at higher gridness values. For simulations with attractors, population gridness, while always lower than mean individual gridness, increased steadily, with 2D exhibiting slightly higher values most of the time, but no substantial increase was observed in the No condition given the absence of alignment between maps. A similar lack of alignment was found for a condition in which recurrent input weights were either too strong or too weak compared with feedforward weights, as well as for a condition in which recurrent inputs to a neuron where shuffled (*Figure 2—figure supplement 1*). The asymptotic behavior for both individual and population gridness was similar for all conditions with attractors. Individual spacing in maps with attractors showed a decrease throughout most of the learning process, more pronounced in the 2D condition, while the No condition evidenced a steady increase toward an asymptote. A combination of progressive increase in gridness and decrease in spacing across days has been observed in animals familiarizing with a novel environment (*Barry et al., 2007*). Our results, although only qualitatively exhibiting similar trends, point to the efficiency of excitatory collaterals in imposing constraints to the population activity as a possible mechanism. This compression of maps resulting from experience, also observable by turning off the attractor in a trained network (*Figure 2—figure supplement 2*), was less evident in the mean population spacing, obtained from average autocorrelograms, indicating that, at least in our simulations, this phenomenon has a strong driver in the deviation of individual cells from the coordinated population behavior, which would explain why the contraction is more marked for the most rigid constraint (2D). Despite these subtle differences, gridness, spacing and spread looked overall very similar across conditions with attractors, and markedly different in the No condition.

## Toroidal topology of the population activity space

Classical features such as gridness and spacing looked similar in maps obtained with different attractor geometry. We next asked, more generally, if the topology of the population activity was also the same for different conditions. Every pixel in the arena where the virtual animal runs is associated with a vector containing the mean activity of each neuron in that position. These vectors are the columns of the population matrix M, where the element $M_{ij}$ is the mean activity of the $i^{th}$ neuron in the $j^{th}$ pixel. This set of vectors form a point cloud of size equal to the number of pixels in a space of dimension $N_{EC}$ (the number of grid cells). It is commonly understood, given the symmetry of grid maps, that this cloud should be a sample of a low dimensional structure represented by a twisted torus and embedded in a high dimensional space (*Knierim and Zhang, 2012*), as recently shown in experimental data (*Gardner et al., 2022*; see Methods and *Figure 3—figure supplement 1*).

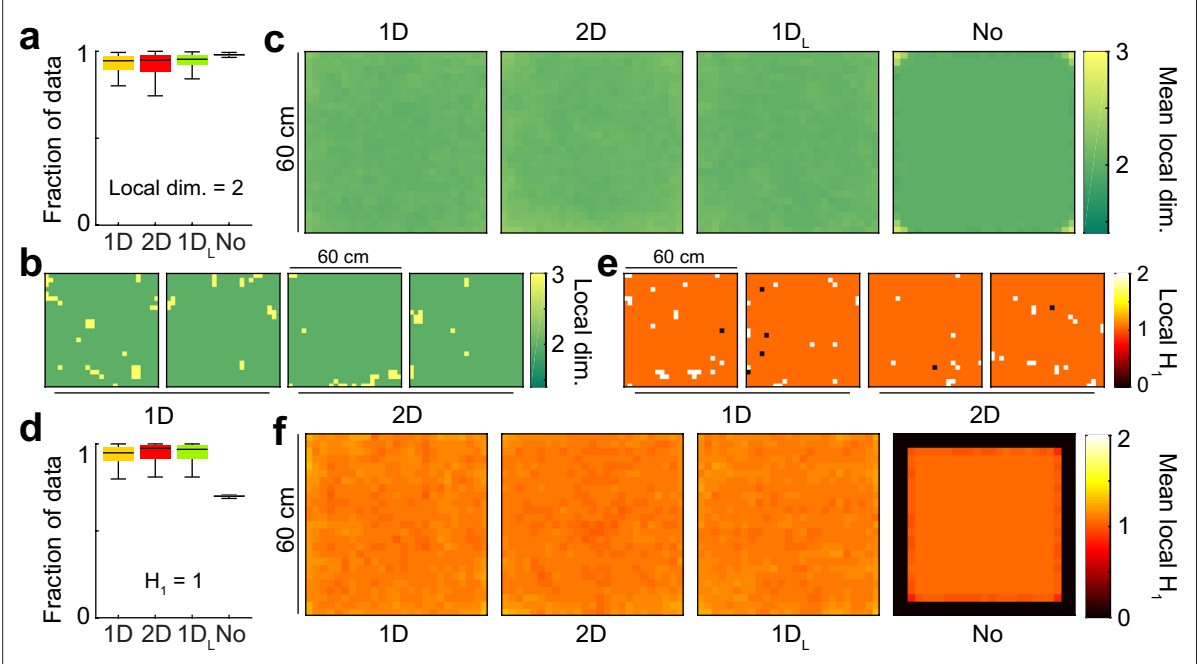

**Figure 3.** The population activity in all attractor conditions is locally two-dimensional, with no boundary or singularities. (**a**) Distribution across conditions of the fraction of the data with local dimensionality of 2 (i.q.r., n = 100). (**b**) Distribution of local dimensionality across physical space in representative examples of 1D and 2D conditions. (**c**) Average distribution of local dimensionality for all conditions (same color code as in (**b**)). (**d-f**) As (**a–c**) for but exploring deviations of the local homology $H_1$ from a value of 1, the value expected away from borders and singularities.

The online version of this article includes the following figure supplement(s) for figure 3:

**Figure supplement 1.** Methods in topological data analysis and examples.

The topology of a point cloud can in some cases be completely determined through a series of methods in topological data analysis. The theorem of classification of closed surfaces states that if the population activity space is a sample of a geometric object that (i) is locally two-dimensional, (ii) has neither boundary nor singularities and (iii) is orientable, then its topology is determined by its homology (*Hatcher, 2002*; see Methods).

To understand whether and when this is true in our simulations, we first studied the local dimension of the population activity space (i) for the different conditions. To avoid irregularities sometimes found at the perimeter of the environment, we used for all further analyses the 60 cm wide central square of each 1 m wide map, and our conclusions apply to this area. For every data point, we extracted the principal components of a local neighborhood around it (*Figure 3—figure supplement 1*). We defined the local dimension at this point as the number of principal components for which an elbow in the rate of explained variance in the local neighborhood was found. In all conditions, most of the data points had a local dimensionality equal to 2 (more than 90% in all conditions), with eventual outliers that had some impact on the mean but not on the median (*Figure 3a*). To understand if these deviations were the result of noise or, in contrast, had a structure in the physical space, we next plotted individual maps of local dimensionality (*Figure 3b*) and their average across simulations of the same condition (*Figure 3c*). We only observed a structure in the No condition, where mean values close to 3 were concentrated at the corners of the reduced map. These results suggest that the population activity in all conditions with attractors is concentrated around a structure with a local dimension of 2.

Next, to understand if the data had boundary or singularities (ii), we studied the local homology of the underlying space by estimating the first Betti number $\beta_1$ in an annulus neighborhood around each data point (*Stolz et al., 2020*; *Figure 3d*). Roughly, Betti numbers ($\beta_0$, $\beta_1$, $\beta_2$) indicate the number of connected components ($\beta_0$), holes ($\beta_1$), or voids ($\beta_2$) in a point cloud. These numbers are estimated from persistence diagrams, which aim to identify cycles in the point cloud that persist across a wide range of typical distances (see Methods and Appendix I). In a sample of a locally two-dimensional manifold with neither boundary nor singularities, the data inside the annulus neighborhood is expected to form

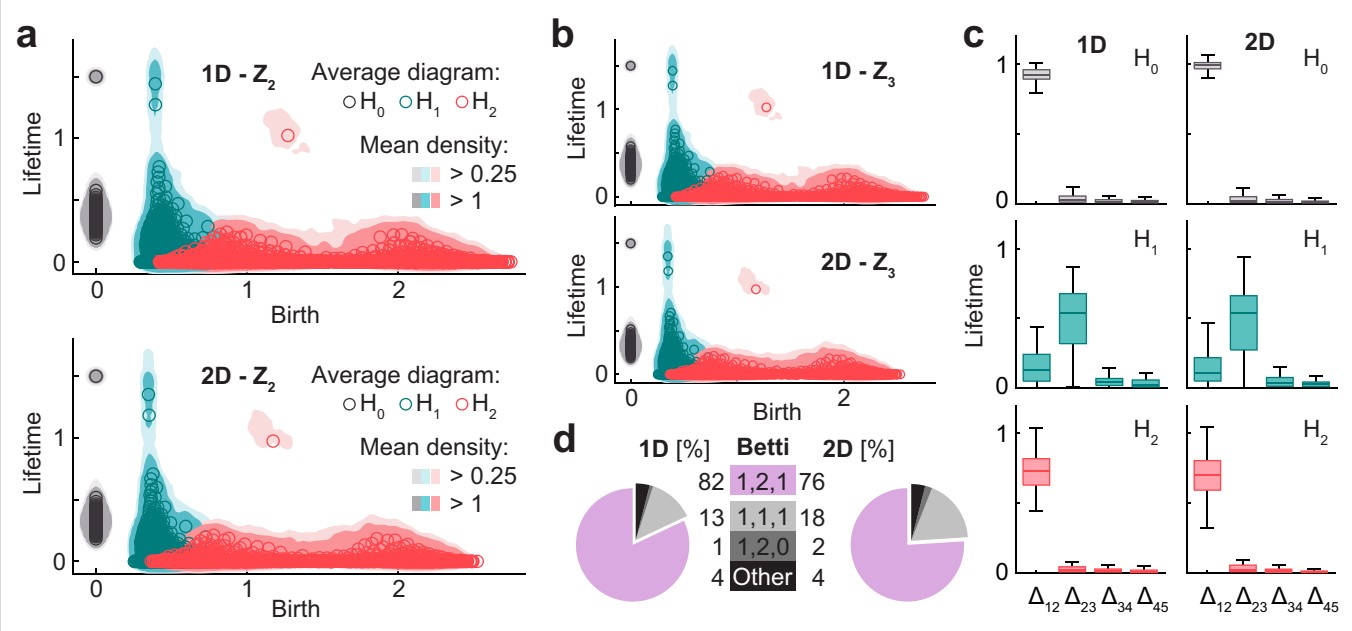

**Figure 4.** For 1D and 2D architectures, the population activity is an orientable manifold with the homology of torus. (**a**) Smoothed density of pooled data (colored areas) and average across simulations (circles) of persistence diagrams with coefficients in $Z_2$ for $H_0$ (grey), $H_1$ (blue), and $H_2$ (red) corresponding to simulations in the 1D (top) and 2D (bottom) conditions. (**b**) As (**a**) but with coefficients in $Z_3$. Similarity with (**a**) implies that population activity lies within an orientable surface. (**c**) Distribution (i.q.r., n = 100) across simulations of lifetime difference between consecutive generators ordered for each simulation from longest to shortest lifetime for 1D (left) and 2D (right). Maxima coincide with Betti numbers. (**d**) Pie plot and table indicating the number of simulations (out of 100 in each condition) classified according to their Betti numbers.

The online version of this article includes the following figure supplement(s) for figure 4:

**Figure supplement 1.** Persistent homology and Betti numbers for condition 1 $_{DL}$.

a ring, with local $\beta_1 = 1$, while points in the boundary are characterized by $\beta_1 = 0$ and singularities by $\beta_1 > 1$ (*Figure 3—figure supplement 1d*). We observed that most of the data points had $\beta_1 = 1$, which was the case for more than 90% of data in conditions with attractors and around 70% in the No condition. The No condition had not only the lowest fraction of data with $\beta_1 = 1$, but also the lowest deviation in the distribution, pointing to a systematic decrease in $\beta_1$. This could be explained by examining individual maps of local $\beta_1$ (*Figure 3e*) and averages across simulations (*Figure 3f*). The conditions with attractors exhibited no structure in eventual deviations from $\beta_1 = 1$, but the average for the No condition had a value $\beta_1 = 0$ in the pixels close to the perimeter of the reduced map, indicating a non-empty boundary set in the population activity.

Put together, these results suggest that the population activity for simulations with attractors is locally two-dimensional, without boundary or singularities, forming a closed surface. In contrast, data clouds in the No condition do not meet the first two conditions of the theorem of classification of closed surfaces, and are compatible with a two-dimensional sheet exhibiting a boundary along the edge of the space selected for analysis.

We next studied the orientability (iii) of the population activity space for the conditions with attractors. For each simulation, we obtained and compared persistence diagrams in $Z_2$ and $Z_3$. The pooled distribution of such diagrams and the averages across 100 simulations of each type (Methods) were almost identical for conditions 1D, 2D (*Figure 4a and b*) and 1 $_{DL}$ (*Figure 4—figure supplement 1*), indicating that the population activity in all cases is an orientable manifold.

Given that for simulations with attractors the three conditions posed by the theorem of classification of closed surfaces were met, we were able to conclude that the topology of the population activity is determined by its Betti numbers. From the average diagrams for all conditions with attractors, Betti numbers could be qualitatively estimated as those of a torus: $\beta_0 = 1$, $\beta_1 = 2$ and $\beta_2 = 1$ (*Figure 4a*). This was the case not only for average persistence diagrams, but also for most of the individual simulations, as shown in the plots of the distribution across simulations of the difference in

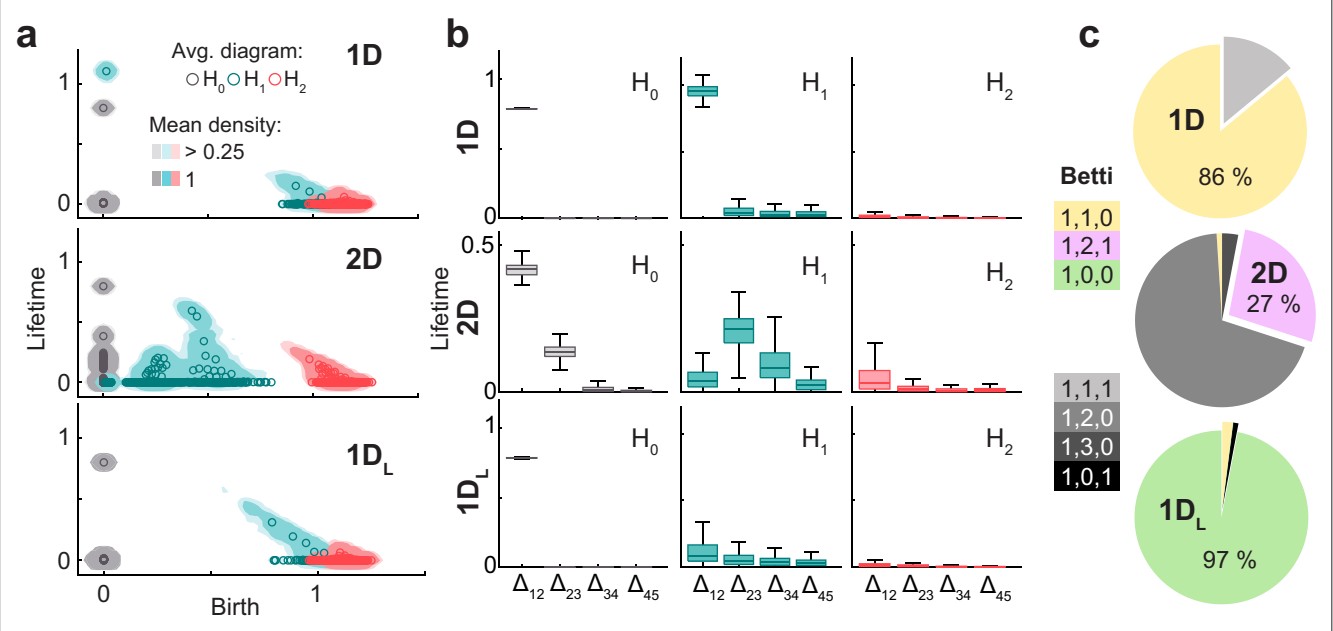

**Figure 5.** Features of the attractor architecture observed in the persistent homology of neural activity. (**a**) From top to bottom, diagrams as in *Figure 4a* but for the point cloud of neurons in conditions 1D, 2D, and 1 $_{DL}$. Each panel shows smoothed density of pooled data (colored areas) and average (circles) persistence diagrams for $H_0$ (gray), $H_1$ (blue), and $H_2$ (red) with coefficients in $Z_2$. (**b**) As in *Figure 4c*, distribution of the difference between consecutive generators ordered for each simulation from longest to shortest lifetime (i.q.r., n = 100). (**c**) Pie plots showing the percentage of simulations in which each combination of Betti numbers was found.

lifetime of consecutive generators ordered from largest to smallest (*Figure 4c*). When quantifying the Betti numbers for individual simulations using a common cutoff value (Methods), 82 (out of 100) were classified as [1, 2, 1] in the 1D condition and 76 in the 2D condition (*Figure 4d*), with small deviations compatible with a noisy scenario for the rest of the simulations.

In summary, our analyses show that different attractor architectures (torus, ring, stripe) similarly constrain the population activity into a torus embedded in a high dimensional space. This is not surprising for the 2D condition, where the architecture has itself the topology of a torus, but is an unprecedented result in the case of the 1D and 1 $_{DL}$ conditions, given that these simpler attractors were not tailored to represent two-dimensional neighboring relations.

## Features of network architecture in the spatial maps

Our analyses showed that all conditions with attractors had a population activity with the topology of a torus, irrespective of the architecture of recurrent connections. We next asked if similar tools could be used to unveil differences between conditions, following the intuition that the network architecture could perhaps be reflected in the geometry of map similarity relationships across neurons. We computed Betti numbers for the point cloud given by spatial maps of individual neurons, represented by a set of $N_{EC} = 100$ points (the number of neurons) in a 625-dimensional space (the number of pixels in the reduced map; *Figure 5*). This is equivalent to what was done in the previous section, but with the transpose of the matrix M introduced there.

For the 2D condition, the mean diagram could qualitatively be described as having the Betti numbers of a torus [1,2,1], which was also apparent in the difference between consecutive ordered lifetimes (*Figure 5a and b*). However, in individual persistence diagrams only 27 simulations had Betti numbers [1,2,1], while 69 had Betti numbers [1,2,0] where the cavity could not be correctly identified (*Figure 5c*). The reason for this discrepancy, or the failure to find the cavity in individual diagrams, is possibly related to the low number of datapoints used in this analysis (100 vs 625 in the previous case) taking the signal-to-noise ratio close to the limit of no detection. In contrast, 1D simulations had an average persistence diagram similar to the majority of individual diagrams (86%), characterized by the

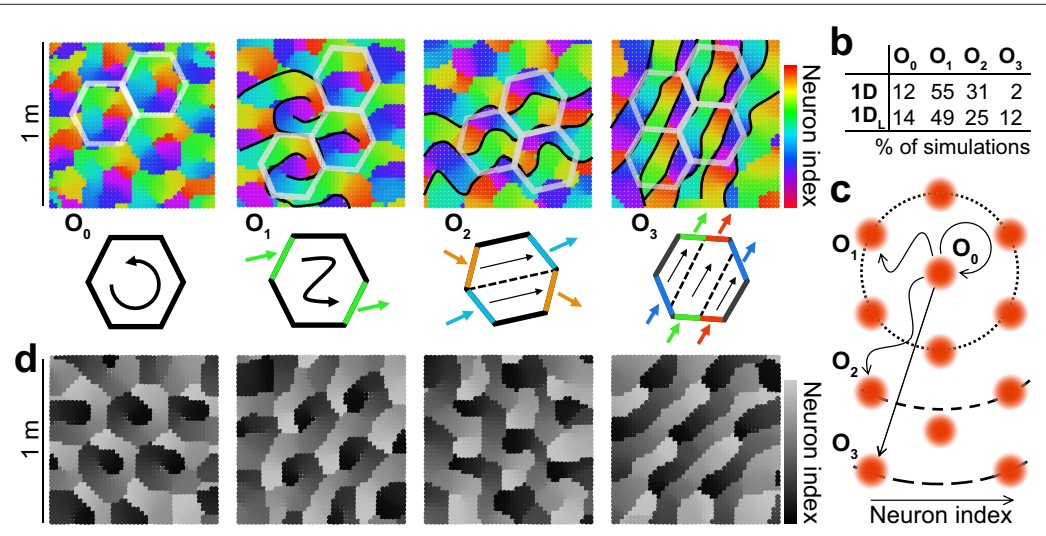

**Figure 6.** Multiple configurations for the alignment of grid maps by one-dimensional attractors. (**a**) Top: Population map for 4 representative examples of 1D simulations with increasing configuration order (indicated) from left to right. Color indicates the region of the ring attractor best describing the mean population activity at each position. Schematics of some of the frontiers, defined by abrupt changes in color (black), and hexagonal tiles maximally coinciding with these frontiers (semi-transparent white) are included for visualization purposes. Bottom: Schematic representation of the order of the solution as the minimum number of colors traversing the perimeter of the hexagonal tile or, equivalently, the minimum number of hexagonal tiles whose perimeter is traversed by one cycle of the attractor. (**b**) Table indicating the number of simulations out of 100 in which each configuration order was found. (**c**) Schematic representation of the ring attractor extending in space from a starting point to different order neighbors in a hexagonal arrangement. (**d**) As (**a**) but for the $1_{DL}$ condition. Gray scale emphasizes the lack of connections between extremes of the stripe attractor.

Betti numbers of a ring [1,1,0], while for $1_{DL}$ simulations the average diagram and 97% of individual diagrams had Betti numbers [1,0,0] compatible with a stripe.

In summary, the analysis on the matrix M showed that all conditions with attractors had a population activity embedded in a torus, while the analysis on its transpose showed qualitative differences across conditions, where in general the homology recapitulated that of the architecture of recurrent connections.

## Flexibility of one-dimensional attractors

Given that one-dimensional attractors are not constructed in an ad hoc way to guarantee the correct organization of the population activity into a pre-defined configuration, we next asked what kind of geometrical arrangement was found by our self-organizing model to allow covering two-dimensional space with a one-dimensional arrangement of neurons. To visualize the population activity in space, we colored neurons in the 1D ring attractor according to hue, so that connected neurons along the ring were assigned similar colors. We then assigned to each pixel in the virtual space the color that best described the mean population activity associated with it. This allowed us to plot for each simulation a map in which color represented the mean population activity (*Figure 6a*). While all individual colors in these maps had hexagonal periodicity, as expected from a population of aligned grid maps, the geometry of the attractor layout in physical space allowed for a classification into 4 orders ($O_0$ to $O_3$) with different prevalence (*Figure 6b*). A way to understand these configurations is to imagine a cycle along the attractor. The cycle begins and ends in the same color, but since in physical space any given color is constrained to have hexagonal periodicity, the end has to lie either in the same place as it started ($O_0$) or in an n-order neighboring field (in our case n=1–3; *Figure 6c*). This conceptualization implies that, although we only found 4 configurations for aligning grid cells with a 1D attractor, the constraint imposed by hexagonal symmetry is compatible with a countably infinite number of them (as many as orders of neighbors in a hexagonal grid), provided attractors are able to stretch enough in physical space. We speculate that the number of neurons in the grid cell layer ($N_{EC}$) could play a

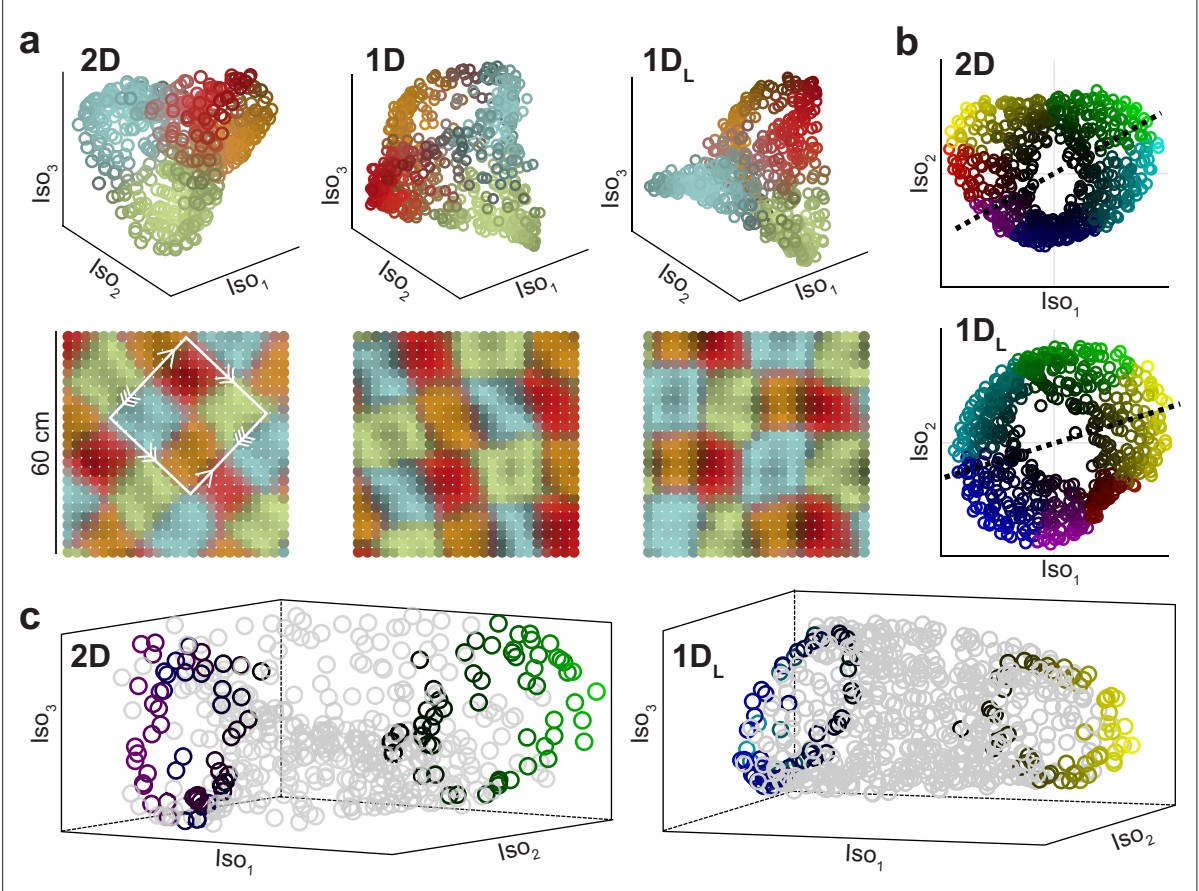

**Figure 7.** Two different visualizations of the twisted torus obtained with Isomap. (**a**) Top: Representative examples of dimensionality reduction into a structure resembling a tetrahedron for different conditions (indicated). Data points are colored according to the distance to one of the four cluster centers obtained with k-means (each one close to tetrahedron vertices; k=4). Bottom: Same data and color code but plotted in physical space. The white square indicates the minimal tile containing all colors, with correspondence between edges, indicated by arrows, matching the basic representation of the twisted torus. (**b**) Representative examples of dimensionality reduction into a torus structure squeezed at two points, obtained in other simulations using identical Isomap parametrization. Hue (color) and value (from black to bright) indicate angular and radial cylindrical coordinates, respectively. (**c**) For the same two examples (indicated), three-dimensional renderings. To improve visualization of the torus cavity, color is only preserved for data falling along the corresponding dashed lines in (**b**).

critical role in determining to what extent the attractor can stretch and which configurations can be achieved in practice by a given network. Simulations of the $1_{DL}$ condition were classified into categories following the same logic, by assessing the distance between the two extremes of the attractor (colored black and white) in terms of neighboring order in a hexagonal grid (**Figure 6b and d**).

These results show that the weak constraint imposed by one-dimensional flexible attractors allows for many possible solutions to co-exist as local minima of the self-organization energy landscape.

## Visualization of the twisted torus

Dimensionality reduction techniques are a popular way of visualizing high dimensional data, such as the population activity in our simulations. It should be noted, however, that in general they provide no guarantee of preserving the topology of the data cloud in the original high dimensional space. For our data, three-dimensional Isomap projections (**Tenenbaum et al., 2000**) allowed for the visualization of the twisted torus in all conditions with attractors. In many simulations, the three-dimensional reduction of the population activity looked like a tetrahedron (**Figure 7a**). If data were grouped using k-means clustering (k=4) in the reduced Isomap space, a four-color checkerboard emerged in physical space. The minimal tile containing all four colors was a square in which one pair of opposite sides had simple periodicity while the other had a periodicity with a 180° twist, which is the basic representation for

the twisted torus (*Figure 3—figure supplement 1*). Other times the same reduction technique, with identical parameters, produced directly a torus-like shape, squeezed at two opposite sides (*Figure 7b and c*). While extreme solutions looked like the tetrahedron or the torus, intermediate visualizations were also found which could not be clearly interpreted as one or the other. The fact that the same procedure produced so different visualizations calls for a cautious approach to interpreting the geometry of reduced data when using non-linear methods such as Isomap. This technique does not aim to preserve the global geometry of the original point cloud, but instead the relative distance between data points.

Our results show that there are multiple ways in which continuous attractors can align grid cells, including simple architectures such as ring or stripe attractors. In topological terms, the resulting population activity is equivalent (homeomorphic), despite differences in the topology of the architecture or in projections obtained through dimensionality-reduction techniques.

## Discussion

Our main result is that the alignment of hexagonal axes in a model of grid cells can result from interactions between neurons following a simple one-dimensional architecture, not constructed ad hoc for representation of two-dimensional spaces. This possibility has not been assessed before in modeling because a common assumption is that recurrent collateral architecture perfectly defines the geometry of the manifold that the population activity is constrained to by the attractor. We show for the first time that this seemingly reasonable assumption is wrong, providing two counter-examples in which the representational space is a torus but the architecture, either a ring or a stripe, has a different topology and even lower dimensionality. Crucially, if the attractor is inactive, weak or shuffled, grid maps obtained under otherwise identical conditions exhibit markedly lower levels of hexagonal symmetry and do not align, failing to constrain the population activity into a torus. Our results open the way to considering the potential of simple flexible attractors for a wide spectrum of modeling applications, given their capability of enacting on the population activity a negotiated constraint, as opposed to rigid attractors with no such degrees of freedom.

Advantages of flexible attractors are versatility and simplicity in network design. Grid cells, for example, represent multiple geometries with individual characteristics in each case. Two-dimensional maps of familiar environments are highly symmetric and periodic, but this is not the case for maps in other dimensionalities. In one- and three-dimensional spatial tasks grid cells exhibit multiple fields, as in two-dimensional navigation, but with larger and more irregular spacing (*Yoon et al., 2016*; *Hafting et al., 2008*; *Grieves et al., 2021*; *Ginosar et al., 2021*). In other one-dimensional tasks, involving the representation of frequency of a sound or time, they much more often develop single response fields (*Aronov et al., 2017*; *Kraus et al., 2015*). To properly model this collection of scenarios with rigid attractors, one should consider a number of them embedded on the same network architecture, each specialized for a single purpose, and possibly a mechanism to select the attractor best suited for every situation. Alternatively, the same could perhaps be achieved with a single architectonic principle. One-dimensional attractors are simple enough to emerge independently of experience, as exhibited by head direction cells in rats prior to eye-opening (*Bjerknes et al., 2015*) or internally generated sequences in the hippocampus (*Pastalkova et al., 2008*). Future computational explorations should include a wider range of architectures to assess whether even simpler configurations than the ones used here, such as a collection of short fragmented sequences, could align grid cells in a similar way.

Most grid cell models, in contrast to the one in this work, are focused on path integration, or the capacity of spatial maps to persist in the absence of spatial inputs based on self-motion information (*Burak and Fiete, 2009*; *Burgess et al., 2007*; *Widloski and Fiete, 2014*). Experiments in which animals navigate in the dark support this functionality for hippocampal and entorhinal maps, and it has been recently shown that the path integrator can be recalibrated in an almost online fashion (*Jayakumar et al., 2019*). However, although from a theoretical perspective grid cells are ideal candidates to implement path integration, the involvement of some or all grid cells in this operation still needs to find direct experimental proof. In contrast, a growing corpus of evidence suggests that grid cells can exhibit behaviors that deviate from pure path integration. This includes local and global distortions of the two-dimensional grid map in response to a variety of experimental manipulations (*Barry et al., 2007*; *Krupic et al., 2015*; *Krupic et al., 2018*; *Boccara et al., 2019*; *Butler et al., 2019*; *Sanguinetti-Scheck and Brecht, 2020*), as well as the progressive refinement in symmetry and decrease in spacing

observed across days of familiarization to a novel environment (*Yoon et al., 2013*; *Barry et al., 2012*). In our model, this last result could be understood as an increase in the efficiency with which collateral connections impose their constraint. As feedforward synaptic weights are modified, neurons become better tuned to the constraint and individual deviation from the collective behavior decreases. More generally, understanding how this heterogeneous set of experimental results could emerge from inter-actions between path integration and mapping is a challenge for future work in which the concept of flexible attractors could prove useful. Path integration is an operation that needs to be computed in the direction of movement, which is at a given instant a one-dimensional space. Many grid cell models can be thought of as employing several overlapping two-dimensional attractors, each specialized in one direction of movement, to achieve path integration in all directions (*Burak and Fiete, 2009*), a task that, we speculate, one-dimensional attractors might be naturally suited for without loss of flexibility.

In a recent experiment, mice were trained to run head-fixed at a free pace on a rotating wheel, in complete darkness and with no other sensory or behavioral feedback (*Cogno et al., 2022*). It could be expected that in such a situation the population activity deprived of inputs is influenced to a greater extent by its internal dynamics, so that this kind of experiment offers a window into the architecture of the attractor. The experimenters observed that the entorhinal population dynamic engaged in cycles, with a period of tens of seconds to minutes. These cycles, naturally occurring here but not in other areas of the brain, point to the possibility of one-dimensional attractor arrangements, modelled by either our 1D or 1 $_{DL}$ conditions, as a prevailing organizational principle of the entorhinal cortex. Future efforts should focus on whether or not a relationship exists between the organization of grid cells within entorhinal cycles and their relative spatial phases in two-dimensional open field experiments, contrasting the population map with configurations shown in *Figure 6*.

Grid cells were originally described in superficial layers of the medial entorhinal cortex, but were later found also in other entorhinal layers and even in an increasing number of other brain areas (*Boccara et al., 2010*; *Long et al., 2021a*; *Long and Zhang, 2021b*). Our work points to the possi-bility that organizational principles simpler than previously thought could act in some of these areas to structure grid cell population activity. In addition, given that grid cell properties change substantially during the early life of rodents (*Langston et al., 2010*; *Wills et al., 2010*), flexible attractors could also be taken into account as a potential intermediate stage toward the formation of more complex architectures.

Our work shows that attractor networks have capabilities that so far have not been exploited in modeling. Addressing the dimensionality of an attractor network, as is common practice to describe it, becomes challenging from the perspective of our results, given that a single network architecture can organize population activity into manifolds of diverse geometry, and the same geometry can be achieved by architectures of different dimensionality. Generally speaking, operations of cross-dimensional embedding achieved by flexible attractors could shed light on the way we map a world of unknown complexity through our one-dimensional experience.

## Methods

The model is inspired in a previous work that describes extensively the mechanism and reasons why Hebbian learning sculpts hexagonal maps through self-organization (*Kropff and Treves, 2008*). We here describe the main ingredients of the model and small modifications aimed to make it simpler and computationally less expensive.

The network has an input layer of $N_I = 225$ neurons projecting to a layer of $N_{EC} = 100$ cells. While the model works with arbitrary spatially stable inputs (*Kropff and Treves, 2008*), for simplicity we use place cell like inputs. Input cells had Gaussian response fields with a standard deviation of 5.4 cm centered at preferred positions uniformly distributed across the 1 m arena.

The total field h received by grid cell i at time t is given by two terms. The first one includes the contributions of the feedforward connections from input cells. The second one includes recurrent contributions

$$h_i\left(t\right) = \sum_{j=1}^{N_I} W_{i,j}^I \, r_j^I\left(t\right) + \sum_{k=1}^{N_{EC}} W_{i,k}^{EC} \, r_k^{EC}\left(t\right)$$

where $r_j^I(t)$ and $r_k^{EC}(t)$ are the firing rate of input cell j and grid cell k, respectively. The feedforward synaptic weight matrix $W^I$ is equipped with Hebbian plasticity, while for the purposes of this paper the recurrent synaptic weigh $W^{EC}$ is fixed (see next section).

The field of the cell is inserted into a set of two equations with two internal variables, $h^{act}$ and $h^{inact}$ and a parameter β aimed to mimic adaptation or neural fatigue within the cell,

$$h_i^{act}(t+1) = h_i(t) - h_i^{inact}(t)$$
$$h_i^{inact}(t+1) = h_i(t) + \beta h_i^{act}(t).$$

Once the value of $h^{act}$ is obtained for all grid cells, a threshold linear transfer function with gain G is applied. A threshold T mimicking inhibition is established so that only the fraction A of cells with highest $h^{act}$ values has non-zero firing rate, while a normalization, acting as an effective gain, ensures that Hebbian plasticity does not get stuck at the beginning of the learning process due to low postsynaptic activity. The activity of each cell is obtained as

$$r_i^{EC}(t+1) = G \frac{\left| h_{act}(t) - T \right|_{>0}}{\left\langle \left| h_{act}(t) - T \right|_{>0} \right\rangle},$$

where the operation $|\ldots|_{>0}$ represents a rectifying linear transformation and <…>denotes averaging across all cells. This normalization is effectively equivalent to controlling the sparseness of the network (*Kropff and Treves, 2008*) but is much more efficient computationally.

The update of the feedforward synaptic weight $W_{i,j}^I$ is given by de Hebbian rule

$$W_{i,j}^I(t+1) = W_{i,j}^I(t) + \varepsilon \left( r_j^I(t) r_i^{EC}(t) - \overline{r_j^I}\, \overline{r_i^{EC}} \right)$$

where ε is a learning parameter and the computationally efficient temporal average operation

$$\overline{r_i}(t+1) = \overline{r_i}(t)(1-\delta) + r_i(t)\delta$$

is used. Negative values of $W^I$ are set to zero and the vector of all presynaptic weights entering a given postsynaptic grid cell is divided by its Euclidean norm to ensure that it remains inside a hypersphere.

In our hands, the condition 2D was the one with the greatest sensibility to model parameters. For this reason, they were fine-tuned using the 2D recurrent architecture, aiming to reduce as much as possible the number of cells and thus optimize the computational cost. This was achieved by reducing the grid cell layer to 100 cells, a number below which self-organization of the population activity into a torus ceased to be consistent. We noticed that including a greater number of neurons in the input layer had a substantial impact on the speed and stability of the learning process, which led us to include 225 input cells. Once the 2D architecture simulations were optimized, the other conditions were run using the same values for all parameters and initial conditions, except for the parameters describing the recurrent collateral architecture itself. The following are some important model parameters.

Parameters ensuring that the mean contributions of feed forward and recurrent inputs to a neuron are of the same order of magnitude:

- Gain A for otherwise normalized recurrent inputs: 2.
- Gain G for feedforward inputs: 0.1.
- Peak value for inputs $r^I$: 20.
- Adaptation parameter β: 0.04.
- Average parameter δ: 0.5
- Side of the square arena: 1 m.
- Input field standard deviation: 5.4 cm.
- Distance traveled in one simulation step: 0.6 cm.
- Variation in direction at each step: normal distribution with 0° mean and 17° s.d.
- Overall number of steps per simulation $2\ 10^7$.

- Grid cells allowed to have non-zero activity at any given time: 60%.

Other parameters:

- Adaptation parameter β: 0.04.

- Average parameter δ: 0.5
- Side of the square arena: 1 m.
- Input field standard deviation: 5.4 cm.
- Distance traveled in one simulation step: 0.6 cm.
- Variation in direction at each step: normal distribution with 0° mean and 17° s.d.
- Overall number of steps per simulation 2 $10^7$.

## Recurrent collateral architectures

### Toroidal architecture

For the purpose of designing the 2D architecture of recurrent collaterals, each neuron in a given simulation was assigned a position, uniformly covering a 2D arena. The strength of connectivity between a given pair of cells k and l was set to depend on their relative position $\mathbf{x} = [x_k - x_l, y_k - y_l]$, through a function f($\mathbf{x}$) that was defined as the sum of three cosine functions in directions $\mathbf{k_i}$, 120° and 240° from each other, i.e. an ideal grid map (**Kropff and Treves, 2008**),

$$f(\boldsymbol{x}) = 1 + \frac{2}{3} \sum_{i=1}^{3} \cos\left(\boldsymbol{k_i x}\right)$$

The spacing of this imaginary grid map (inverse to the modulus of $\mathbf{k}$) could be varied along a wide range of values without noticeable consequences on the simulations. For simulations included in this work it was set to 60 cm.

### Ring architecture

For the 1D condition, neurons were uniformly distributed along an imaginary ring, spaced by 3.6°. The connection strength between any pair of neurons was defined as proportional to a 7.2° standard deviation Gaussian function of the minimum angle between them.

### Stripe architecture

For the 1 $_{DL}$ condition, neurons were uniformly distributed along an imaginary stripe. The connection strength between any pair of neurons was defined as proportional to Gaussian function of the distance between them, with standard deviation equal to twice the distance between consecutive neurons.

### Fragmented architecture

To prove that characteristic organization of spatial phases is not a necessary outcome of flexible attractors, a Fragmented 1D condition was used (**Figure 1—figure supplement 1**). The architecture of connectivity was constructed by repeating 20 times the process of randomly selecting 10 cells and adding to their mutual weights those corresponding to a 1 $_{DL}$ attractor connecting them. The resulting architecture corresponds to the overlap of 20 short 1 $_{DL}$ attractors. Such an architecture can be understood as simpler to obtain from biological processes compared to other 1D architectures studied here, but more difficult to fully characterize, which led us to restrict the analysis of this architecture to demonstrating that flexible attractors do not necessarily require organized spatial phases to align grid cells.

## Rate maps

Mean rate for each pixel in space was obtained from the instantaneous rate of each neuron observed during visits to the pixel. To optimize memory usage, at any given time the pixel currently traversed by the rat was identified and its mean rate for each neuron j, $m_j$ updated as

$$m_j = m_j \left(1 - \tau\right) + r_j \left(t\right) \tau$$

where $r_j$ is the instantaneous firing rate and $\tau$ is 0.03. The rest of the pixels of the map were not modified at this step.

Autocorrelograms were obtained by correlating two copies of each map displaced relatively to one another in all directions and magnitudes. To reduce the absolute value close to the borders, where correlations can reach extreme values with poor statistical power, a 1 m circular hamming window was

applied to each autocorrelogram. Mean population autocorrelograms were obtained by averaging the autocorrelogram across all neurons in a given simulation.

## Quantification of grid properties

### Spacing

Autocorrelograms were interpolated to a Cartesian grid in polar coordinates, so that correlation could be analyzed for all angles at any fixed radius. Spacing was defined as the radius with maximal 6-period Fourier component modulation of the correlation across angles.

### Gridness

For the radius that defined spacing, gridness was defined as the mean autocorrelation at the six 60 degree spaced maxima minus that at the six 60 degree spaced minima.

### Angular spread

For each cell in a given simulation, the six maxima around the center of the autocorrelogram were identified. A k-means clustering algorithm was applied to the pool of all maxima in the simulation (MATLAB *kmeans*() function, with k=6, otherwise default parameters and 10 repetitions to avoid local minima). The spread was defined as the mean absolute angle difference between pairs of points belonging to the same cluster.

## Metric structure of the population activity

To study the topology of the population activity (*Figure 4*), the central 60 cm of each map in a simulation was considered. The population activity thus determines point clouds of 625 points — the size of the arena, that is, 25x25 — in $R^{N_{EC}}$, where $N_{EC} = 100$ is the number of simulated grid cells. For the purpose of capturing the intrinsic geometry of the underlying space determined by these point clouds, and to avoid the effects of the 'curse of dimensionality', we endowed each point cloud with an estimator of the *geodesic distance*. This estimator, known as the *kNN-distance*, is defined as the length of the shortest path along the *k-nearest neighbors graph*, a graph with an edge between every data point and each of its k-nearest neighbors with respect to the ambient Euclidean distance. We set the value k=10 for all our analyses but similar results could be obtained for a range of similar values of k.

To recover network architecture features (*Figure 5*), we studied the simultaneous spatial activity of grid cells. The associated point cloud is a set of $N_{EC} = 100$ vectors in $R^{625}$ representing the average activity of every grid cell on each pixel of the arena. This point cloud, when endowed with the metric structure given by the Pearson correlation distance, shares geometric features with the combinatorial architecture of the underlying neural network.

## Persistent homology

We aimed to robustly recover geometric information, such as the number of connected components, cycles and holes of different dimensions, from the simulated data (*Figure 3—figure supplement 1a*). To do so, we computed the *persistent homology Boissonnat et al., 2018*; *Edelsbrunner and Harer, 2008*; *Edelsbrunner et al., 2024*; *Zomorodian and Carlsson, 2005* of each point cloud endowed with its respective metric structure. As output we obtained *persistence diagrams*, graphical representations of the evolution of the generators of the *homology groups* associated to each point cloud, for different parameter scales (*Figure 3—figure supplement 1c*). Each generator is described as a point whose first coordinate represents its *birth* and the second coordinate, its *lifetime*. Generators with long lifetime indicate topological features, while the ones with short lifetime are linked to noise. Note that persistent homology at degree 0 encodes the evolution of connected components. It is always the case that a single generator of $H_0$ has an infinite lifetime, as a consequence of the compactness of the point cloud. Its lifetime was set to an arbitrary value larger than generators associated to noise but with a similar magnitude, to facilitate visualization. To summarize the information of the persistent homology over *all* simulations, we computed both the average of the persistence diagrams as its *Frechet mean* (*Mileyko et al., 2011*; *Turner et al., 2014*), as well as the density associated with the distribution of points in the diagrams (*Figure 4*).

All the computations of persistence diagrams, related to both the population activity and the cross-cell similarities, were performed with the Ripser package (**Bauer, 2021**), while the Frechet mean (or barycenter of persistence diagrams) was obtained using the corresponding library in GUDHI (**Maria et al., 2024**).

## Automated quantification of individual persistence diagrams

Betti numbers are typically assessed from persistence diagrams in a qualitative way. Profiting from the fact that we had 100 similar persistence diagrams for every condition, we designed an automated procedure to determine cutoff values for each homology group and condition. A histogram of lifetime with 100 bins between 0 and the maximum value was obtained for the pool of all cycles in all simulations belonging to the condition. The histogram was smoothed with a 3-bin standard deviation gaussian window. Locations of minima in this smoothed histogram were identified and the one representing the greatest fall from the previous maximum was set as the cutoff lifetime value. Persistence diagrams for individual simulations were analyzed by counting how many cycles had a lifetime greater than the corresponding cutoff value.

## Local principal component analysis

Local *Principal Component Analysis (PCA)* is a well-established procedure to detect the local dimension of point clouds (**Fukunaga and Olsen, 1971**). It is based on the popular method PCA of linear dimensionality reduction, applied to local k-nearest neighborhoods of each data point (**Figure 3—figure supplement 1b**). We employed local neighborhoods of size k=70 for all simulations of population activity with attractors and k=20 for the ones in the No condition. These values were determined as the center of a range of k values with stable outcomes.

For every local neighborhood, we computed the evolution of the *rate of explained variance* after adding each principal component (in decreasing eigenvalue-order). An estimator of the local dimension at a point is the number of dimensions at which there is a drop off (or 'elbow') in the curve of explained variances (**Figure 3—figure supplement 1b**). For elbow detection we used the Python package kneed **Satopaa, 2011**.

## Local persistent homology

Persistent homology can also be used to capture the topological structure *around* each data point. Even though homology does not distinguish among local neighborhoods of different dimensions (and hence, it is not useful to identify local dimensions), it is an appropriate method to detect anomalies such as points in the *boundary* or *singularities*. The main idea is to identify the shape of the region *surrounding* each point, by studying the persistent homology of an annular neighborhood (**Stolz et al., 2020**).

We defined the *annular local neighborhood* of a point x as the set of points in the point cloud (ordered according to the Euclidean distance to x) between the $k_1^{th}$ and the $k_2^{th}$ nearest neighbors, with $k_1 = 50$, $k_2 = 100$ for the simulations with attractor, and $k_1 = 10$, $k_2 = 30$ for the ones in the No condition (**Figure 3—figure supplement 1d**). We used the Ripser package **Bauer, 2021** for the computation of local persistent homology.

## Orientability

Orientability is the geometric property that ensures a consistent local coordinate system in a manifold. In the special case of *closed* manifolds (compact connected manifolds without boundary), this homeomorphism invariant can be detected by its homology.

We computed the persistence diagrams of the point clouds obtained from 100 simulations of the population activity of grid cells in all conditions, using coefficients in both $Z_2$ and $Z_3$ (**Figure 3—figure supplement 1e**). A summary of the persistent homology over all the simulations (for every coefficient field) was presented via the Frechet mean and the density of the distribution of the generators in the persistent diagrams.

Note that for *any* closed manifold M of dimension 2, $H_2(M, Z_2) \neq 0$. This is consistent with salient generator in the (Frechet mean) persistence diagram for $H_2$ that we can detect in all conditions with attractors (**Figure 4**). We also observe that the Frechet mean of persistence diagrams remains unaltered after the change of coefficients from $Z_2$ to $Z_3$. This proves the orientability of the underlying

surfaces in all cases. If the sample belonged to a non-orientable surface, the salient generator of the persistent diagram representing $H_2(M, Z_2)$ should disappear when compared with $H_2(M, Z_3)$ (*Figure 3—figure supplement 1e*). This should also be accompanied by the disappearance of a salient generator of $H_1(M, Z_2)$ when contrasted with $H_1(M, Z_3)$. This phenomenon of simultaneous changes in homology is explained by the independence of the Euler characteristic on the choice of field of coefficients.

## Dimensionality reduction

Among the most popular techniques in manifold learning are the procedures for *dimensionality reduction*, that aim to project high-dimensional point clouds into a low-dimensional space while preserving some properties of the original data.

*Isomap* is a celebrated (non-linear) dimensionality reduction method that assumes the data is a sample of a low dimensional manifold embedded in a high dimensional Euclidean space. It reduces the dimensionality by mapping the original data into a lower dimensional Euclidean space while preserving geodesic distances on the manifold subjacent in the data. Since the intrinsic distance in the underlying manifold is unknown, it estimates the geodesic distance by the kNN graph distance, where the parameter k represents the number of nearest neighbors used in the construction of the graph.

We performed Isomap projections of the population activity in all conditions 2D, 1D, 1 $_{DL}$ and No condition, with a parameter value of k=10 (although comparable results are obtained for a range of similar values). We employed the method Isomap from the Python library sklearn.manifold.

We remark that, even though dimensionality reduction procedures may serve as a useful tool for data visualization and feature extraction as part of a machine learning pipeline, they do not provide guarantee a priori to preserve the topology of the underlying manifold, so they do not constitute in general a proof of structure of the original data neither an accurate preprocessing method for a subsequent rigorous geometric analysis.

## Acknowledgements

This work was supported by Human Frontiers Science Program grant RGY0072/2018 (EK), Argentina Foncyt grant PICT 2019–2596 (EK) and EPSRC grant EP/R018472/1 (XF).

## Additional information

### Competing interests

Emilio Kropff: Reviewing editor, *eLife*. The other authors declare that no competing interests exist.

### Funding

| Funder | Grant reference number | Author |
|---|---|---|
| Human Frontier Science Program | RGY0072/2018 | Emilio Kropff |
| Agencia Nacional de Promoción de la Investigación, el Desarrollo Tecnológico y la Innovación | PICT 2019-2596 | Emilio Kropff |
| EPSRC | EP/R018472/1 | Ximena Fernandez |

The funders had no role in study design, data collection and interpretation, or the decision to submit the work for publication.

### Author contributions

Sabrina Benas, Formal analysis, Investigation, Writing - original draft; Ximena Fernandez, Conceptualization, Formal analysis, Methodology, Writing - original draft; Emilio Kropff, Conceptualization, Resources, Formal analysis, Funding acquisition, Investigation, Visualization, Methodology, Writing - original draft, Project administration, Writing - review and editing

## Author ORCIDs
Emilio Kropff ![ORCID] https://orcid.org/0000-0001-5996-8436

Reviewer #1 (Public Review): https://doi.org/10.7554/eLife.89851.3.sa1
Reviewer #3 (Public Review): https://doi.org/10.7554/eLife.89851.3.sa2
Author response https://doi.org/10.7554/eLife.89851.3.sa3

## Additional files

### Supplementary files
• MDAR checklist

### Data availability
All data have been made publicly available in GitHub (copy archived at *Benas, 2024*).

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
