## [Editor Report · eLife assessment]

In this **valuable** study, the authors use a computational model to investigate how recurrent connections influence the firing patterns of grid cells, which are thought to play a role in encoding an animal's position in space. The work suggests that a one-dimensional network architecture may be sufficient to generate the hexagonal firing patterns of grid cells, a possible alternative to attractor models based on recurrent connectivity between grid cells. However, the support for this proposal was **incomplete**, as some conclusions for how well the model dynamics are necessary to generate features of grid cell organization were not well supported.

---

## [Referee Report · Reviewer #1 (Public Review)]

I'll begin by summarizing what I understand from the results presented, and where relevant how my understanding seems to differ from the authors' claims. I'll then make specific comments with respect to points raised in my previous review (below), using the same numbering. Because this is a revision I'll try to restrict comments here to the changes made, which provide some clarification, but leave many issues incompletely addressed.

As I understand it the main new result here is that certain recurrent network architectures promote emergence of coordinated grid firing patterns in a model previously introduced by Kropff and Treves (Hippocampus, 2008). The previous work very nicely showed that single neurons that receive stable spatial input could 'learn' to generate grid representations by combining a plasticity rule with firing rate adaptation. The previous study also showed that when multiple neurons were synaptically connected their grid representations could develop a shared orientation, although with the recurrent connectivity previously used this substantially reduced the grid scores of many of the neurons. The advance here is to show that if the initial recurrent connectivity is consistent with that of a line attractor then the network does a much better job of establishing grid firing patterns with shared orientation.

Beyond this point, things become potentially confusing. As I understand it now, the important influence of the recurrent dynamics is in establishing the shared orientation and not in its online generation. This is clear from Figure S3, but not from an initial read of the abstract or main text. This result is consistent with Kropff and Treves' initial suggestion that 'a strong collateral connection... from neuron A to neuron B... favors the two neurons to have close-by fields... Summing all possible contributions would result in a field for neuron B that is a ring around the field of neuron A.' This should be the case for the recurrent connections now considered, but the evidence provided doesn't convincingly show that attractor dynamics of the circuit are a necessary condition for this to arise. My general suggestion for the authors is to remove these kind of claims and to keep their interpretations more closely aligned with what the results show.

Major (numbered according to previous review)

(1) Does the network maintain attractor dynamics after training? Results now show that 'in a trained network without feedforward Hebbian learning the removal of recurrent collaterals results in a slight increase in gridness and spacing'. This clearly implies that the recurrent collaterals are not required for online generation of the grid patterns. This point needs to be abundantly clear in the abstract and main text so the reader can appreciate that the recurrent dynamics are important specifically during learning.

(2) Additional controls for Figure 2 to test that it is connectivity rather than attractor dynamics (e.g. drawing weights from Gaussian or exponential distributions). The authors provide one additional control based on shuffling weights. However, this is far from exhaustive and it seems difficult on this basis to conclude that it is specifically the attractor dynamics that drive the emergence of coordinated grid firing.

(3) What happens if recurrent connections are turned off? The new data clearly show that the recurrent connections are not required for online grid firing, but this is not clear from the abstract and is hard to appreciate from the main text.

(4) This is addressed, although the legend to Fig. S2D could provide an explanation / definition for the y-axis values.

(5) Given the 2D structure of the network input it perhaps isn't surprising that the network generates 2D representations and this may have little to do with its 1D connectivity. The finding that the networks maintain coordinated grids when recurrent connections are switched off supports my initial concern and the authors explanation, to me at least, remain confusing. I think it would be helpful to consider that the connectivity is specifically important for establishing the coordinated grid firing, but that the online network does not require attractor dynamics to generate coordinated grid firing.

(6) Clarity of the introduction. This is somewhat clearer, but I wonder if it would be hard for someone not familiar with the literature to accurately appreciate the key points.

(7) Remapping. I'm not sure why this is ill posed. It seems the proposed model can not account for remapping results (e.g. Fyhn et al. 2007). Perhaps the authors could just clearly state this as a limitation of the model (or show that it can do this).

Previous review:

This study investigates the impact of recurrent connections on grid fields generated in networks trained by adjusting the strength of feedforward spatial inputs. The main result is that if the recurrent connections in the network are given a 1D continuous attractor architecture, then aligned grid firing patterns emerge in the network following training. Detailed analyses of the low dimensional dynamics of the resulting networks are then presented. The simulations and analyses appear carefully carried out.

The feedforward model investigated by the authors (previously introduced by Kropff & Treves, 2008) is an interesting and important alternative to models that generate grid firing patterns through 2-dimensional continuous attractor network (CAN) dynamics. However, while both classes of model generate grid fields, in making comparisons the manuscript is insufficiently clear about their differences. In particular, in the CAN models grid firing is a direct result of their 2-D architecture, either a torus structure with a single activity bump (e.g. Guanella et al. 2007, Pastoll et al. 2013), or sheet with multiple local activity bumps (Fuhs & Touretzky, Burak & Fiete, 2009). In these models, spatial input can anchor the grid representations but is not necessary for grid firing. By contrast, in the feedforward models neurons transform existing spatial inputs into a grid representation. Thus, the two classes of model implement different computations; CANs path integrate, while the feedforward models transform spatial representations. A demonstration that a 1D CAN generates coordinated 2D grid fields would be surprising and important, but its less clear why coordination between grids generated by the feedforward mechanism would be surprising. As written, it's unclear which of these claims the study is trying to make. If the former, then the conclusion doesn't appear well supported by the data as presented, if the latter then the results are perhaps not so unexpected, and the imposed attractor dynamics may still not be relevant.

Whichever claim is being made, it could be helpful to more carefully evaluate the model dynamics given predictions expected for the different classes of model. Key questions that are not answered by the manuscript include:

- At what point is the 1D attractor architecture playing a role in the models presented here? Is it important specifically for training or is it also contributing to computation in the fully trained network?

- Is an attractor architecture required at all for emergence of population alignment and gridness? Key controls missing from Figure 2 include training on networks with other architectures. For example, one might consider various architectures with randomly structured connectivity (e.g. drawing weights from exponential or Gaussian distributions).

- In the trained models do the recurrent connections substantially influence activity in the test conditions? Or after training are the 1D dynamics drowned out by feedforward inputs?

- What is the low dimensional structure of the input to the network? Can the apparent discrepancy between dimensionality of architecture and representation be resolved by considering structure of the inputs, e.g. if the input is a 2 dimensional representation of location then is it surprising that the output is too?

- What happens to representations in the trained networks presented when place cells remap? Is the 1D manifold maintained as expected for CAN models, or does it reorganise?

---

## [Referee Report · Reviewer #3 (Public Review)]

Summary:

The paper proposes an alternative to the attractor hypothesis, as an explanation for the fact that grid cell population activity patterns (within a module) span a toroidal manifold. The proposal is based on a class of models that were extensively studied in the past, in which grid cells are driven by synaptic inputs from place cells in the hippocampus. The synapses are updated according to a Hebbian plasticity rule. Combined with an adaptation mechanism, this leads to patterning of the inputs from place cells to grid cells such that the spatial activity patterns are organized as an array of localized firing fields with hexagonal order. I refer to these models below as feedforward models.

It has already been shown by Si, Kropff, and Treves in 2012 that recurrent connections between grid cells can lead to alignment of their spatial response patterns. This idea was revisited by Urdapilleta, Si, and Treves in 2017. Thus, it should already be clear that in such models, the population activity pattern spans a manifold with toroidal topology. The main new contributions in the present paper are (i) in considering a form of recurrent connectivity that was not directly addressed before. (ii) in applying topological analysis to simulations of the model. (iii) in interpreting the results as a potential explanation for the observations of Gardner et al.

Strengths:

The exploration of learning in a feedforward model, when recurrent connectivity in the grid cell layer is structured in a ring topology, is interesting. The insight that this not only align the grid cells in a common direction but also creates a correspondence between their intrinsic coordinate (in terms of the ring-like recurrent connectivity) and their tuning on the torus is interesting as well, and the paper as a whole may influence future theoretical thinking on the mechanisms giving rise to the properties of grid cells.

Weaknesses:

(1) In Si, Kropff and Treves (2012) recurrent connectivity was dependent on the head direction tuning, in addition to the location on a 2d plane, and therefore involved a ring structure. Urdapilleta, Si, and Treves considered connectivity that depends on the distance on a 2d plane. The novelty here is that the initial connectivity is structured uniquely according to latent coordinates residing on a ring.

(2) The paper refers to the initial connectivity within the grid cell layer as one that produces an attractor. However, it is not shown that this connectivity, on its own, indeed sustains persistent attractor states. Furthermore, it is not clear whether this is even necessary to obtain the results of the model. It seems possible that (possibly weaker) connections with ring topology, that do not produce attractor dynamics but induce correlations between neurons with similar locations on the ring would be sufficient to align the spatial response patterns during the learning of feedforward weights.

(3) Given that all the grid cells are driven by an input from place cells that span a 2d manifold, and that the activity in the grid cell network settles on a steady state which is uniquely determined by the inputs, it is expected that the manifold of activity states in the grid cell layer, corresponding to inputs that locally span a 2d surface, would also locally span a 2d plane. The result is not surprising. My understanding is that this result is derived as a prerequisite for the topological analysis, and it is therefore quite technical.

(4) The modeling is all done in planar 2d environments, where the feedforward learning mechanism promotes the emergence of a hexagonal pattern in the single neuron tuning curve. Under the scenario in which grid cell responses are aligned (i.e. all neurons develop spatial patterns with the same spacing and orientation) it is already quite clear, even without any topological analysis that the emerging topology of the population activity is a torus.

However, the toroidal topology of grid cells in reality has been observed by Gardner et al also in the wagon wheel environment, in sleep, and close to boundaries (whereas here the analysis is restricted to the a sub-region of the environment, far away from the walls). There is substantial evidence based on pairwise correlations that it persists also in various other situations, in which the spatial response pattern is not a hexagonal firing pattern. It is not clear that the mechanism proposed in the present paper would generate toroidal topology of the population activity in more complex environments. In fact, it seems likely that it will not do so, and this is not explored in the manuscript.

(5) Moreover, the recent work of Gardner et al. demonstrated much more than the preservation of the topology in the different environments and in sleep: the toroidal tuning curves of individual neurons remained the same in different environments. Previous works, that analyzed pairwise correlations under hippocampal inactivation and various other manipulations, also pointed towards the same conclusion. Thus, the same population activity patterns are expressed in many different conditions. In the present model, this preservation across environments is not expected. Moreover, the results of Figure 6 suggest that even across distinct rectangular environments, toroidal tuning curves will not be preserved, because there are multiple possible arrangements of the phases on the torus which emerge in different simulations.

(6) In real grid cells, there is a dense and fairly uniform representation of all phases (see the toroidal tuning of grid cells measured by Gardner et al). Thus, the highly clustered phases obtained in the model (Fig. S1) seem incompatible with the experimental reality. I suspect that this may be related to the difficulty in identifying the topology of a torus in persistent homology analysis based on the transpose of the matrix M.

(7) The motivations stated in the introduction came across to me as weak. As now acknolwledged in the manuscript, attractor models can be fully compatible with distortions of the hexagonal spatial response patterns - they become incompatible with this spatial distortions only if one adopts a highly naive and implausible hypothesis that the attractor state is updated only by path integration. While attractor models are compatible with distortions of the spatial response pattern, it is very difficult to explain why the population activity patterns are tightly preserved across multiple conditions without a rigid two-dimentional attractor structure. This strong prediction of attractor models withstood many experimental tests - in fact, I am not aware of any data set where substantial distortions of the toroidal activity manifold were observed, despite many attempts to challenge the model. This is the main motivation for attractor models. The present model does not explain these features, yet it also does not directly offer an explanation for distortions in the spatial response pattern.

(8). There is also some weakness in the mathematical description of the dynamics. Mathematical equations are formulated in discrete time steps, without a clear interpretation in terms of biophysically relevant time scales. It appears that there are no terms in the dynamics associated with an intrinsic time scale of the neurons or the synapses (a leak time constant and/or synaptic time constants). I generally favor simple models without lots of complexity, yet within this style of modelling, the formulation adopted in this manuscript is unconventional, introducing a difficulty in interpreting synaptic weights as being weak or strong, and a difficulty in interpreting the model in the context of other studies.

In my view, the weaknesses discussed above limit the ability of the model, as it stands, to offer a compelling explanation for the toroidal topology of grid cell population activity patterns, and especially the rigidity of the manifold across environments and behavioral states. Still, the work offers an interesting way of thinking on how the toroidal topology might emerge.

---

## [Author Response]

The following is the authors' response to the current reviews.

**Reviewer #1 (Public Review):**
I'll begin by summarizing what I understand from the results presented, and where relevant how my understanding seems to differ from the authors' claims. I'll then make specific comments with respect to points raised in my previous review (below), using the same numbering. Because this is a revision I'll try to restrict comments here to the changes made, which provide some clarification, but leave many issues incompletely addressed.As I understand it the main new result here is that certain recurrent network architectures promote emergence of coordinated grid firing patterns in a model previously introduced by Kropff and Treves (Hippocampus, 2008). The previous work very nicely showed that single neurons that receive stable spatial input could 'learn' to generate grid representations by combining a plasticity rule with firing rate adaptation. The previous study also showed that when multiple neurons were synaptically connected their grid representations could develop a shared orientation, although with the recurrent connectivity previously used this substantially reduced the grid scores of many of the neurons. The advance here is to show that if the initial recurrent connectivity is consistent with that of a line attractor then the network does a much better job of establishing grid firing patterns with shared orientation.Beyond this point, things become potentially confusing. As I understand it now, the important influence of the recurrent dynamics is in establishing the shared orientation and not in its online generation. This is clear from Figure S3, but not from an initial read of the abstract or main text. This result is consistent with Kropff and Treves' initial suggestion that 'a strong collateral connection... from neuron A to neuron B... favors the two neurons to have close-by fields... Summing all possible contributions would result in a field for neuron B that is a ring around the field of neuron A.' This should be the case for the recurrent connections now considered, but the evidence provided doesn't convincingly show that attractor dynamics of the circuit are a necessary condition for this to arise. My general suggestion for the authors is to remove these kind of claims and to keep their interpretations more closely aligned with what the results show.

We would like to clarify that the simple (flexible) attractor is a weaker condition than the ones previously used to align grid cells. However, by no means we claim that it is a necessary condition for grid maps to align. Other architectures, certainly more complex ones but perhaps even simpler ones, can align grid maps in our model.

Major (numbered according to previous review)(1) Does the network maintain attractor dynamics after training? Results now show that 'in a trained network without feedforward Hebbian learning the removal of recurrent collaterals results in a slight increase in gridness and spacing'. This clearly implies that the recurrent collaterals are not required for online generation of the grid patterns. This point needs to be abundantly clear in the abstract and main text so the reader can appreciate that the recurrent dynamics are important specifically during learning.

We respectfully disagree with the interpretation of this result. In this model cells self-organize to produce aligned grid maps. In such systems it makes sense to characterize the equilibrium states of the system. We turned learning off in Figure S3 to show that the recurrent connections have a contractive effect on grid spacing. But artificially turning off learning means that one can no longer make claims about the equilibrium states of the system, since it can no longer evolve freely. In a functional network, if the recurrent attractor is removed, the system will evolve towards poor gridness and no alignment no matter what the starting point is, as also shown in Figure S3. Several experimental results invite us to think of grid cells as the equilibrium solution of a series of constraints that is ready to change at any time: Barry et al, 2012; Yoon et al, 2013; Carpenter et al, 2015; Krupic et al, 2015; Krupic et al, 2018; Jayakumar et al, 2019.

One point in which we perhaps agree with the reviewer is that information about the hexagonal maps is kept in the feedforward weights, while behavior and the recurrent collaterals act as constraints of which these feedforward weights are the equilibrium solution.

(2) Additional controls for Figure 2 to test that it is connectivity rather than attractor dynamics (e.g. drawing weights from Gaussian or exponential distributions). The authors provide one additional control based on shuffling weights. However, this is far from exhaustive and it seems difficult on this basis to conclude that it is specifically the attractor dynamics that drive the emergence of coordinated grid firing.

Again, we do not claim that this is the only way in which grid maps can be aligned, but it is the simplest one proposed so far. We were asked if it was the specific combination of input weights to a cell rather than the organization provided by the attractor which resulted in aligned maps. By shuffling the inputs to a cell we keep the combination of inputs invariant but lose the attractor architecture. Since grid maps in this new situation are not aligned, we can safely conclude that it is not the combination of inputs per se, but the specific organization of these inputs that allows grid alignment. It is not fully clear to us what ‘exhaustive’ means in this context.

(3) What happens if recurrent connections are turned off? The new data clearly show that the recurrent connections are not required for online grid firing, but this is not clear from the abstract and is hard to appreciate from the main text.

This point is related to (1). Absent this constraint, Figure S3 shows that the system evolves toward larger spacing, with poorer gridness and no alignment.

(4) This is addressed, although the legend to Fig. S2D could provide an explanation / definition for the y-axis values.

We have now added: Mean input fields are the sum of all inputs of a given kind entering a neuron at a given moment in time, averaged across cells and time.

(5) Given the 2D structure of the network input it perhaps isn't surprising that the network generates 2D representations and this may have little to do with its 1D connectivity. The finding that the networks maintain coordinated grids when recurrent connections are switched off supports my initial concern and the authors explanation, to me at least, remain confusing. I think it would be helpful to consider that the connectivity is specifically important for establishing the coordinated grid firing, but that the online network does not require attractor dynamics to generate coordinated grid firing.

This point is related to (1) and (3). We agree with the reviewer that the input lies within a 2D manifold, but this is not something that the network has to find out because it receives one datapoint of information at a time. This alone is not enough to form aligned grid cells, since each grid cell can find a roughly equivalent equilibrium in a different direction. It is only the constraint imposed by the recurrent collaterals that aligns grid maps, and, as we show, this constraint does not need to be constructed ad hoc to work on 2D, as previously thought. When recurrent connections are switched off, the system evolves toward unaligned grid maps, with larger spacing and lower gridness.Regarding the results obtained after modifying the network and turning off learning, we think they have a very limited scope (in this case showing the contractive effect of recurrent collaterals on grid spacing), given that the system is artificially being kept out of its natural equilibrium.

(6) Clarity of the introduction. This is somewhat clearer, but I wonder if it would be hard for someone not familiar with the literature to accurately appreciate the key points.

We have made our best effort to improve the clarity of the introduction.

(7) Remapping. I'm not sure why this is ill posed. It seems the proposed model can not account for remapping results (e.g. Fyhn et al. 2007). Perhaps the authors could just clearly state this as a limitation of the model (or show that it can do this).

We view our model as perfectly consistent with Fyhn et al, 2007. Remapping is not triggered by the network itself, though, but rather by a re-arrangement of the inputs requiring the network to learn new associations. Different simulations of the same model with identical parameters can be interpreted as remapping experiments.

**Reviewer #3 (Public Review):**
Summary:The paper proposes an alternative to the attractor hypothesis, as an explanation for the fact that grid cell population activity patterns (within a module) span a toroidal manifold. The proposal is based on a class of models that were extensively studied in the past, in which grid cells are driven by synaptic inputs from place cells in the hippocampus. The synapses are updated according to a Hebbian plasticity rule. Combined with an adaptation mechanism, this leads to patterning of the inputs from place cells to grid cells such that the spatial activity patterns are organized as an array of localized firing fields with hexagonal order. I refer to these models below as feedforward models.It has already been shown by Si, Kropff, and Treves in 2012 that recurrent connections between grid cells can lead to alignment of their spatial response patterns. This idea was revisited by Urdapilleta, Si, and Treves in 2017. Thus, it should already be clear that in such models, the population activity pattern spans a manifold with toroidal topology. The main new contributions in the present paper are (i) in considering a form of recurrent connectivity that was not directly addressed before. (ii) in applying topological analysis to simulations of the model. (iii) in interpreting the results as a potential explanation for the observations of Gardner et al.

We wanted to note that we do not see this paper as proposing an alternative to the attractor hypothesis, given that we use attractor networks, but rather as an exploration of possibilities not yet visited by this hypothesis.

Strengths:The exploration of learning in a feedforward model, when recurrent connectivity in the grid cell layer is structured in a ring topology, is interesting. The insight that this not only align the grid cells in a common direction but also creates a correspondence between their intrinsic coordinate (in terms of the ring-like recurrent connectivity) and their tuning on the torus is interesting as well, and the paper as a whole may influence future theoretical thinking on the mechanisms giving rise to the properties of grid cells.Weaknesses:(1) In Si, Kropff and Treves (2012) recurrent connectivity was dependent on the head direction tuning, in addition to the location on a 2d plane, and therefore involved a ring structure. Urdapilleta, Si, and Treves considered connectivity that depends on the distance on a 2d plane. The novelty here is that the initial connectivity is structured uniquely according to latent coordinates residing on a ring.

The recurrent architectures in the cited works are complex and require arranging cells in a 2D manifold to calculate connectivity based on their relative 2D position. In other words, the 2D structure is imprinted in the architecture, as in our 2D condition. In this work the network is much simpler and only requires neighboring relations in 1D. Such relationships have been shown to spontaneously emerge in the hippocampal formation (Pastalkova et al, 2008; Gonzalo Cogno et al, 2024).

(2) The paper refers to the initial connectivity within the grid cell layer as one that produces an attractor. However, it is not shown that this connectivity, on its own, indeed sustains persistent attractor states. Furthermore, it is not clear whether this is even necessary to obtain the results of the model. It seems possible that (possibly weaker) connections with ring topology, that do not produce attractor dynamics but induce correlations between neurons with similar locations on the ring would be sufficient to align the spatial response patterns during the learning of feedforward weights.

Regarding the first part of the comment, the recurrent collaterals create one or at times multiple bumps of activity in the network so that neighboring (interconnected) cells activate together. An initial random state of activity rapidly falls into this dynamic, constrained by the attractor. To us this is not surprising given that this connectivity is the classical means of creating a continuous attractor. Perhaps there is some deeper meaning in this comment that we are not fully grasping.

Regarding the second part of the comment, we fully agree with the reviewer. We are presenting what so far is the simplest connectivity that can align grid maps, but by no means we claim that it is the simplest possible one. Regarding weaker connections with ring topology, we show in Figure S2 that a ring attractor with too weak or too strong connections is incapable of aligning grids, since a balance between feedforward and feedback inputs is required.

(3) Given that all the grid cells are driven by an input from place cells that span a 2d manifold, and that the activity in the grid cell network settles on a steady state which is uniquely determined by the inputs, it is expected that the manifold of activity states in the grid cell layer, corresponding to inputs that locally span a 2d surface, would also locally span a 2d plane. The result is not surprising. My understanding is that this result is derived as a prerequisite for the topological analysis, and it is therefore quite technical.

We understand that the reviewer is referring to the motivation behind studying local dimensionality. We agree that the topological analysis approach is quite technical, but it provides unique insights. The theorem of closed surfaces, which allows us to deduce a toroidal topology from Betti numbers (1,2,1), only applies to closed surfaces. One thus needs to show that the point cloud is a surface (local dimensionality of 2) and is closed (no borders or singularities). If borders or singularities were present, a toroidal topology could not be claimed from these Betti numbers. Thus, it is a crucial step of the analysis.

(4) The modeling is all done in planar 2d environments, where the feedforward learning mechanism promotes the emergence of a hexagonal pattern in the single neuron tuning curve. Under the scenario in which grid cell responses are aligned (i.e. all neurons develop spatial patterns with the same spacing and orientation) it is already quite clear, even without any topological analysis that the emerging topology of the population activity is a torus.However, the toroidal topology of grid cells in reality has been observed by Gardner et al also in the wagon wheel environment, in sleep, and close to boundaries (whereas here the analysis is restricted to the a sub-region of the environment, far away from the walls). There is substantial evidence based on pairwise correlations that it persists also in various other situations, in which the spatial response pattern is not a hexagonal firing pattern. It is not clear that the mechanism proposed in the present paper would generate toroidal topology of the population activity in more complex environments. In fact, it seems likely that it will not do so, and this is not explored in the manuscript.

We agree that our work was constrained to exploration in 2D and that the situations posed by the reviewer are challenging, but we do not see them as unsurmountable. The wagon wheel shows a preservation of toroidal topology locally, where the behavior of the animal is rather 2-dimensional. Globally, hexagonal maps are lost, which is compatible with some flexibility in the way grid maps are formed. If sleep meant that all inputs are turned off, our model would predict a dynamic dictated by the architecture (1D for the ring attractor, for example), but we do not really know that this is the case. In the future, we intend to explore predictive activity along the linear attractor, which could both result in path integration and in some level of preservation of the activity when inputs are completely turned off.

Regarding boundaries, as we have argued before, the cited work chooses to filter away what looks like more than half of the overall explained variance through PCA, and this is only before applying a non-linear dimensionality reduction algorithm. It is specifically shown that the analyzed components are the ones with global periodicity throughout the environment. Thus, it is conceivable that through this approach, local irregularities found only at the borders are disregarded in favor of a clearer global picture. While using a different methodology, our approach follows a similar spirit, albeit with far less noisy data.

(5) Moreover, the recent work of Gardner et al. demonstrated much more than the preservation of the topology in the different environments and in sleep: the toroidal tuning curves of individual neurons remained the same in different environments. Previous works, that analyzed pairwise correlations under hippocampal inactivation and various other manipulations, also pointed towards the same conclusion. Thus, the same population activity patterns are expressed in many different conditions. In the present model, this preservation across environments is not expected. Moreover, the results of Figure 6 suggest that even across distinct rectangular environments, toroidal tuning curves will not be preserved, because there are multiple possible arrangements of the phases on the torus which emerge in different simulations.

We agree with this observation. A symmetry in our implementation results in the fact that only ~50% of times the system falls in the preferred solution, and the rest of the times it falls into other local minima. Whether this result is at odds with current observations can be debated on the basis of probabilities. However, we believe that the symmetry we found is purely circumstantial, and that it can be broken by elements such as head direction modulation or other ingredients used to achieve path integration. In other words, we acknowledge that symmetry is an issue of the implementation we show here (which has been kept as simple as possible to serve as a proof-of-principle) but we do not think that it is a defining feature of flexible attractors in general. We expect that future implementations that incorporate path integration capabilities will not present this kind of symmetry in the space of solutions.

Regarding the rigid phase translation across modalities, while this effect is very clear in Gardner et al, it is less so in other datasets. The analyses shown in Hermansen et al (2024) can rather be interpreted as somewhere in the way between perfect rigid translation and fully randomized phases across navigation modalities.

(6) In real grid cells, there is a dense and fairly uniform representation of all phases (see the toroidal tuning of grid cells measured by Gardner et al). Thus, the highly clustered phases obtained in the model (Fig. S1) seem incompatible with the experimental reality. I suspect that this may be related to the difficulty in identifying the topology of a torus in persistent homology analysis based on the transpose of the matrix M.

We partly agree with this observation and note that a pattern of ordered phases is an issue not only for the 1D attractor but also for the 2D one, which appears much more uniform than in experimental data. The low number of neurons we used for computational economy and the full connectivity could be key ingredients to generate these phase patterns. To show that this is not a defining feature of flexible attractors, apart from the fact that these patterns appear also with non-flexible 2D architectures, we included in Figure S1 simulations with ‘fragmented 1D’ architectures. In this case the architecture is a superposition of 20 random 1D stripe-like attractors. While the alignment of maps achieved with this architecture is almost at the same level as the one obtained with 1D and 2D attractors, the phases are much more similar to what has been observed experimentally, and less uniform than what is obtained with 2D attractors.

(7) The motivations stated in the introduction came across to me as weak. As now acknolwledged in the manuscript, attractor models can be fully compatible with distortions of the hexagonal spatial response patterns - they become incompatible with this spatial distortions only if one adopts a highly naive and implausible hypothesis that the attractor state is updated only by path integration. While attractor models are compatible with distortions of the spatial response pattern, it is very difficult to explain why the population activity patterns are tightly preserved across multiple conditions without a rigid two-dimentional attractor structure. This strong prediction of attractor models withstood many experimental tests - in fact, I am not aware of any data set where substantial distortions of the toroidal activity manifold were observed, despite many attempts to challenge the model. This is the main motivation for attractor models. The present model does not explain these features, yet it also does not directly offer an explanation for distortions in the spatial response pattern.

Some interesting examples are experiments in 3D, where grid cells presumably communicate with each other through the same recurrent collaterals, but global periodicity is lost and only some local order is preserved even away from boundaries (Ginosar et al, 2021; Grieves et al, 2021). While these datasets have not been explored using topological analysis, they serve as strong motivators to understanding 2D grid cells as one equilibrium solution that arises under some set of constraints, but belongs to a wider space of possible solutions that may arise as well under more flexible constraints. Even (and especially) if one adheres to the hypothesis that grid cells are pre-wired into a 2D torus, a concept like flexible attractors might become useful to understand how their activity is rendered in 3D. Another strong motivation is our lack of understanding of how a perfectly balanced 2D structure is formed and maintained. Simpler architectures could be thought of as alternatives, but also as an intermediate step towards it.

Regarding the rigid phase translation across modalities, while this effect is very clear in Gardner et al, it is less so in other datasets. The analyses shown in Hermansen et al (2024) can rather be interpreted as somewhere in the way between perfect rigid translation and fully randomized phases.

In a separate point, although it might not be strictly related to the comment, we do not fully share the idea that persistent activity patterns during sleep are necessary or sufficient conditions for attractor dynamics, although we do agree that attractors could be the mechanism behind them and any alternative is at least as complex as attractors. On the necessity side, attractors in the hippocampus are not constantly engaged (Wills et al, 2005). For sufficiency, one should prove that no other network is capable of reproducing the phenomenon, and to our best knowledge we are still far from that point.

(8) There is also some weakness in the mathematical description of the dynamics. Mathematical equations are formulated in discrete time steps, without a clear interpretation in terms of biophysically relevant time scales. It appears that there are no terms in the dynamics associated with an intrinsic time scale of the neurons or the synapses (a leak time constant and/or synaptic time constants). I generally favor simple models without lots of complexity, yet within this style of modelling, the formulation adopted in this manuscript is unconventional, introducing a difficulty in interpreting synaptic weights as being weak or strong, and a difficulty in interpreting the model in the context of other studies.

We chose to keep the model as simple as possible and in the line of previous publications developing it. However, we see the usefulness of putting it in what in the meantime has become a canonical framework. Fortunately this has been done by D’Albis and Kempter (2017). In our simplified version of the model there is no leak term and adaptation on its own brings down activity in the absence of input, but we agree that such a term could be added, albeit not without modifying all other network parameters.

In my view, the weaknesses discussed above limit the ability of the model, as it stands, to offer a compelling explanation for the toroidal topology of grid cell population activity patterns, and especially the rigidity of the manifold across environments and behavioral states. Still, the work offers an interesting way of thinking on how the toroidal topology might emerge.
**Reviewer 1:**

**Reviewer #1 (Recommendations For The Authors):**
See comments above. In addition:(1) Abstract: '...interconnected by a two-dimensional attractor guided by path integration'. This is unclear. I think the intended meaning might be along the lines of '...their being computed by a 2D continous attractor that performs path integration'?'path integration allowing for no deviations from the hexagonal pattern' This is incorrect. Local modulation of the gain of the speed input to a standard CAN would distort the grid pattern.'Using topological data analysis, we show that the resulting population activity is a sample of a torus' Activity in the model?'More generally, our results represent a proof of principle against the intuition that the architecture and the representation manifold of an attractor are topological objects of the same dimensionality, with implications to the study of attractor networks across the brain' I guess one might hold this intuition, but it strikes me as obvious that if you impose an sufficiently strong n-dimensional input on a network then it it's activity could have the same dimensionality. I don't really see this as being a point worth highlighting. Perhaps the more interesting point, it that during learning the recurrent connectivity aligns the grid fields of neurons in the network, and this may be a specific function of the 1D attractor dynamcis, although I don't think the authors have made this point convincing.'The flexibility of this low dimensional attractor allows it to negotiate the geometry of the representation manifold with the feedforward inputs'. See above for comments on the use of 'negotiate'.'while the ensemble of maps preserves features of the network architecture'. I don't understand this. What is the 'ensemble of maps' and what are the features referred to.

We have reviewed the abstract considering these points. Regarding the ‘strong n-dimensional input’, we want to point out that it is not the input itself that generates a torus (the no attractor condition does not lead to a torus) but rather the interplay between the input and the attractor.

‘Perhaps the more interesting point …’, we do not fully understand how this sentence deviates from our own conclusions. We here show that a strong n-dimensional input is not enough to align grid cells (produce a n-torus), it is the interplay between inputs and attractor dynamics that does so, even if the attractor is not n-dimensional in terms of architecture.

The ensemble of maps refers to the transpose of the population activity matrix, where each point in the cloud is a map, and the features refer to the persistent homology.

(2) The manuscript still fails to clarify the difference between a model that path integrates in two dimensions and a model that simply represents information with a given dimensionality. The argument that it's surprising that a network with 1D architecture represents a higher dimensional input strikes me as incorrect and an unnecessary attempt to argue for conceptual importance. At least to me this isn't surprising. It would be surprising if the 1D network could path integrate but this doesn't seem to be the case.

In response to the reviewer’s concerns, we have made clear in the introduction and discussion that this model has no path integration capabilities, although we aim to develop a model capable of path integration using the kind of simple architecture presented here. We want to highlight here that equating attractor dynamics with path integration would be a conceptual mistake.

(3) Other wording also seems to make unnecessary conceptual claims. E.g. The repeated use of 'negotiate' implies some degree of intelligence, or at least an exchange of information, that isn't shown to exist. I wonder if more precise language could be used? As I understand it the dimensionality is bounded by the inputs on the one hand, and the network connectivity on the other, with the actual dimensionality being a function of the recurrent and feedforward synaptic weights. There's clearly some role for the relative weights and the properties of plasticity rules, but I don't see any evidence for a negotiation.

An interesting observation in Figure S2 is that grid maps are aligned only if the relative strength of feedforward and recurrent inputs is similar. If one of them can impose over the other, grid maps do not align. This equilibrium can metaphorically be thought of as a negotiation instance, where the negotiation is an emergent property of the system rather than something happening at an individual synapse.

The following is the authors’ response to the original reviews.

**Reviewer #1:**

**Reviewer #1 (Recommendations For The Authors):**
Major(1) What is the evidence that, after training, the 1D network maintains its attractor dynamics when feedforward inputs are active? If the claim is that it does then it's important to provide evidence, e.g. responses to perturbations, or other tests. The alternative is that after training the recurrent inputs are drowned out by the feed forward spatial inputs.

We agree with the reviewer on the importance of this point. In our model, networks are always learning, and the population activity represented by aligned grid maps in a trained network is a dynamic equilibrium that emerges from the interplay between feedforward and collateral constraints. If Hebbian learning is turned off, one gets a snapshot of the network at that moment. We now show in Fig. S3 that in a trained network without feedforward Hebbian learning the removal of recurrent collaterals results in a slight increase in gridness and spacing. The expansion is due to the fact that, as we argue in the Results section, the attractor has a contractive effect on grid maps, which could relate to observations in novel environments (Barry et al, 2007). If Hebbian learning is turned on in the same situation, the maps, no longer constrained by the attractor, drift toward the equilibrium solution of the ‘No attractor’ condition, with significantly larger spacing, no alignment and lower individual gridness. Thus, the attractor is the force preventing them to do so when feedforward Hebbian learning is on.

These observations point to the key role played by the attractor not only in forming but also in sustaining grid activity. The dynamic equilibrium framework fits well known properties of the system, such as its capacity to recalibrate very fast (Jayakumar et al, 2019), although this particular feature cannot be modeled with the current version of our model, that lacks path integration capabilities.

(2) It would be useful to include additional control conditions for Figure 2 to test the hypothesis that it is simply connectivity, rather than attractor dynamics, that drives alignment.This could be achieved by randomly assigning strengths to the recurrent connections, e.g. drawing from exponential or Gaussian distributions.

We agree and have included Fig. S2b-d, showing that the same distribution of collateral input weights entering each neuron, but lacking the 1D structure provided by the attractor, does not align grid maps. This is achieved by shuffling rows in the connectivity matrix, while avoiding self connections to make the comparison fair (self connections substantially alter the dynamic of the network, making it much more rigid). We observed that individual grid maps have very low gridness levels, even lower than in the no-attractor condition. In contrast, they have levels of population gridness slightly higher than in the no-attractor condition, but closer to 0 than to levels achieved with attractors. Our interpretation of these results is that irregular connectivity achieves some alignment in a few arbitrary directions and/or locations, which improves the coordination between maps at the expense of impairing rather than improving hexagonal responses of individual cells. Such observations stand in clear context to what is observed with continuous attractors with an orderly architecture.

These results suggest that it is the structure of the attractor that allows grid cells to be aligned rather than the mere presence of recurrent collateral connections.

(3) It seems conceivable that once trained the recurrent connections would no longer be required for alignment. Can this be evaluated by considering what happens if the recurrent connections are turned off after training (or slowly turned off during training)? Does the network continue to generate aligned grid fields?

This point has elements in common with point 1. As we argued in that response, the attractor has two main effects on grid maps: it aligns them and it contracts them. If the attractor is turned off, feedforward Hebbian learning progressively drives maps toward the solution obtained for the ‘no attractor’ condition, characterized by maps with larger spacing, poorer gridness and lack of alignment.

(4) After training what is the relative strength of the recurrent and feedforward inputs to each neuron?

Both recurrent and feedforward synaptic-strength matrices are normalized throughout training, so that the overall incoming synaptic strength to each neuron is invariant. Because of this, although individual feed-forward and recurrent input fields vary dynamically, their average is constant, with the exception of the very first instances of the simulation, before a stable regime is reached in grid-cell activity levels. We have included Fig. S2d, showing the dynamics of feedforward and recurrent mean fields throughout learning as well as their ratio. In addition, Fig. S2a shows that the strength of recurrent relative to feedforward inputs is an important parameter, since alignment is only obtained in an intermediate range of ratios.

(5) It would be helpful to also evaluate the low dimensional structure of the input to the network. Assuming it has a 2D structure, as it represents 2D space, can an explanation be provided for why it is surprising that the trained network also encodes activity with a 2D manifold? It strikes me that the more interesting finding might relate to alignment of the grids rather than claims about a 1D attractor encoding a 2D representation. Either way, stronger evidence and clearer discussion would be helpful.

The reviewer is correct in assuming that the input has a 2D structure, that can be represented by a sheet embedded in a high dimensional space and thus has the Betti numbers [1,0,0]. The surprising element in our results is that we are showing for the first time that the population activity of an attractor network is constrained to a manifold that results from the negotiation between the architecture of the attractor and the inputs, and does not merely reflect the former as previously assumed. In this sense, the alignment of grid cells by a 1D attractor is an instance of the more general case that 1D attractors can encode 2D representations.

It is certainly the case that the 2D input is a strong constraint pushing population activity toward a 2D manifold. However, the final form of the 2D manifold is strongly constrained by the attractor, as shown by the contrast with the no-attractor condition (a 2D sheet, as in the input, vs a torus when the attractor is present). The 1D attractor is able to flexibly adapt to the constraint posed by the inputs while doing its job (as demonstrated in previous points), which results in 2D grid maps aligned by a 1D attractor. Generally speaking, this work provides a proof of principle demonstrating that the topology of the attractor architecture and the manifold of the population activity space need not be identical, as previously widely assumed by the attractor community, and need not even have the same dimensionality. Instead, a single architecture can potentially be applied to many purposes. Hence, our work provides a valuable new perspective that applies to the study of attractors throughout the brain.

(6) The introduction should be clearer about the different types of grid model and the computations they implement. E.g. The authors' previous model generates grid fields from spatial inputs, but if my understanding is correct it isn't able to path integrate. By contrast, while the many 2D models with continuous attractor dynamics also generate grid representations, they do so by path integration mechanisms that are computationally distinct from the spatial transformation implemented by feedforward models (see also general comments above).

We agree with the reviewer and have made this point explicit in the introduction.

(7) A prediction from continuous attractor models is that when place cells remap the low dimensional manifold of the grid activity is unaffected, except that the location of the activity bump is moved. It strikes me as important to test whether this is the case for the model presented here (my intuition is that it won't be, but it would be important to establish either way).

We want to emphasize that our model is a continuous attractor model, so the question regarding the difference between what our model and continuous attractor network models predict is an ill-posed one. One of our main conclusions is precisely that attractors can work in a wider spectrum of ways than previously thought.

In lack of a better definition, our multiple simulations could be thought of as training in different arenas. It is true that in our model maps take time to form, but this is also the case in novel environments (Barry et al, 2007), and continuous attractor models exclusively or strongly guided by self motion cues struggle to replicate this phenomenon. We show that the current version of our model accepts multiple solutions (in practice four but conceptually infinite countable), all of them resulting in a torus for the population activity (i.e. the same topology or low dimensional manifold). It is not clear to us how easy it would be to differentiate between most of these solutions in experimental data, with only incomplete information. This said, incorporating a symmetry-breaking ingredient to the model, for example related to head direction modulation, could perhaps lead to the prevalence of a single type of solution. We intend to explore this possibility in the future in order to add path-integration capabilities to the system, as described in the discussion.

(8) The Discussion implies that 1D networks could perform path integration in a manner similar to 2D networks. This is a strong claim but isn't supported by evidence in the study. I suggest either providing evidence that this is the case for models of this kind or replacing it with a more careful discussion of the issue.

The current version of our model has no path integration capabilities, as is now made explicit in the Introduction and Discussion. In addition, we have now made clear that the idea that path integration could perhaps be implemented using 1D networks is, although reasonable, purely speculative.

Minor(1) Introduction. 'direct excitatory communication between them'. Suggest rewording to 'local synaptic interactions', as communication can also be purely inhibitory (e.g. Burak and Fiete, 2009) or indirect by excitation of local interneurons (e.g. Pastoll et al., Neuron, 2013).

We agree and have adopted this phrasing.

(2) The decision to focus the topology analysis on the 60 cm wide central square appears somewhat arbitrary. Are the irregularities referred to a property of the trained networks or would they also emerge with analysis of simulated ideal data? Can more justification be expanded and supplementary analyses be shown when the whole arena is used?

In practical terms, a subsampling of the data to around half was needed because the persistent homology packages struggle to handle large amounts of data, especially in the calculation of H2. We decided to cut a portion of contiguous pixels in the open field at least larger than the hexagonal tile representing the whole grid population period (as represented in Figure 6). Leaving the borders aside was a logical choice since it is known that the solution at the borders is particularly influenced by the speed anisotropy of the virtual rat (see Si, Kropff & Treves, 2012), in a way that mimics how borders locally influence grid maps in actual rats (Krupic et al, 2015). The specific way in which our virtual rat handles borders is arbitrary and might not generalize. A second issue around borders is that maps are differently affected by incomplete smoothing, although this issue does not apply to our data because we did not smooth across neighboring pixels. In sum, considering the central 60 cm wide square was sufficient to contain the whole torus and a reasonable compromise that would allow us to perform all analyses in the part of the environment less influenced by boundaries.

(3) It could help the general reader to briefly explain what a persistence diagram is.

This is developed in the Appendix, but we have now added a reference to it and a brief description in the main text.

(4) For the analyses in Figure 3-4, and separately for Figure 5, it might help the reader to provide visualizations of the low dimensional point cloud.

All these calculations take place in the original high-dimensional point cloud. Doing them in a reduced space would be incorrect because there is no dimensionality reduction technique that guarantees the preservation of topology. In Figure 7 we reduce the dimensionality of data but emphasize that it is only done for visualization purposes, not to characterize topology. We also point out in this Figure that the same non-linear dimensionality reduction technique applied to objects with identical topology yields a wide variety of visualizations, some of them clear and some less clear. This observation further exemplifies why one cannot assume that a dimensionality-reduction technique preserves topology, even for a low-dimensional object embedded in a high-dimensional space.

(5) The detailed comparison of the dynamics of each model is limited by the number of data points. Why not address this by new simulations with more neurons?

We are not sure we understand this comment. In Figure 2, the dynamics for each model are markedly different. These are averages over 100 simulations. We are not sure what benefit would be obtained from adding more neurons. Before starting this work we searched for the minimal number of neurons that would result in convergence to an aligned solution in 2D networks, which we found to be around 100. Optimizing this parameter in advance was important to reduce computational costs throughout our work.

(6) Could the variability in Figure 7 also be addressed by increasing the number of data points?

As we argued in a previous point, there is no reason to expect preservation of topology after applying Isomap. We believe this lack of topology preservation to be the main driver of variability.

(7) Page/line numbers would be useful.

We agree. However, the text is curated by biorxiv which, to our best knowledge, does not include them.

**Reviewer 2:**

**Reviewer #2 (Recommendations For The Authors):**
(1) I highly suggest that the author rewrite some parts of the Results. There are lots of details which should be put into the Methods part, for example, the implementation details of the network, the analysis details of the toroidal topology, etc. It will be better to focus on the results part first in each section, and then introduce some of the key details of achieving these results, to improve the readability of the work.

This suggestion contrasts with that of Reviewer #1. As a compromise, we decided to include in the Results section only methodological details that are key to understanding the conclusions, and describe everything else in the Methods section.

(2) 'Progressive increase in gridness and decrease in spacing across days have been observed in animals familiarizing with a novel environment...' From Fig.2c I didn't see much decrease. The authors may need to carry out some statistical test to prove this. Moreover, even the changes are significant, this might be not the consequence of the excitatory collateral constraint. To prove this, the authors may need to offer some direct evidence.

We agree that the decrease is not evident in this figure due to the scale, so we are adding the correlation in the figure caption as proof. In addition, several arguments, some related to new analyses, demonstrate that the attractor contracts grid maps. First, the ‘no attractor’ condition has a markedly larger spacing compared to all other conditions (Fig. 2a). We also now show that spacing monotonically decreases with the strength of recurrent relative to feedforward weights, in a way that is rather independent of gridness (Fig. S2a). Second, as we now show in Fig. S2b-d, simulations with a shuffled 1D attractor, such that the sum of input synapses to each neuron are the same as in the 1D condition but no structure is present, lead to a spacing that is mid-way between the ‘no attractor’ condition and the conditions with attractors. Third, as we now show in Fig. S3a, turning off both recurrent connections and feedforward learning in a trained network results in a small increase in spacing. Fourth, as we now show in Fig. S3b, turning off recurrent connections while feedforward learning is kept on increases grid spacing to levels comparable to those of the ‘no attractor’ condition. All these elements support a role of the attractor in contracting grid spacing.

(3) Some of the items need to be introduced first before going into details in the paper, for instance, the stipe-like attractor network, the Betti number, etc.

We have added in the Results section a brief description and references to full developments in the Appendix.

**Reviewer 3 (Public Review):**
(1) It is not clear to me that the proposal here is fundamentally new. In Si, Kropff and Treves (2012) recurrent connectivity was dependent on the head direction tuning and thus had a ring structure. Urdapilleta, Si, and Treves considered connectivity that depends on the distance on a 2d plane.

In the work of Si et al connectivity is constructed ad-hoc for conjunctive cells to represent a torus, it depends on head-directionality but also on the distance in a 2D plane. The topology of this architecture has not been assessed, but it is close to the typical 2D ‘rigid’ constraint. In the work of Urdapilleta et al, the network is a simple 2D one. The difference with our work is that we focus on the topology of the recurrent network and do not use head-direction modulation. In this context, we prove that a 1D network is enough to align grid cells and, more generally, we provide a proof of principle that the topology of the architecture and the representation space of an attractor network do not need to be identical, as previously assumed by the attractor community. These two important points were neither argued, speculated nor self-evident from the cited works.

(2) The paper refers to the connectivity within the grid cell layer as an attractor. However, would this connectivity, on its own, indeed sustain persistent attractor states? This is not examined in the paper. Furthermore, is this even necessary to obtain the results in the model? Perhaps weak connections that do not produce an attractor would be sufficient to align the spatial response patterns during the learning of feedforward weights, and reproduce the results? In general, there is no exploration of how the strength of collateral interactions affects the outcome.

The reviewer makes several important points. Local excitation combined with global inhibition is the archetypical architecture for continuous attractors (see for example Knierim and Zhang, Annual review of neuroscience, 2012). Thus, in the absence of feedforward input, we observe a bump of activity. As in all continuous attractors, this bump is not necessarily ‘persistent’ and instead is free to move along the attractor.

We cannot prove that there is not a simpler architecture that has the same effect as our 1D or 1DL conditions, and we think that there are some interesting candidates to investigate in the future. What we now prove in new Fig. S2b-d is that it is not the strength of recurrent connections themselves, but instead the continuous attractor structure that aligns grid cells in our model. To demonstrate this, we shuffle incoming recurrent connections to each neuron in the 1D condition (while avoiding self-connections for fairness), and show that training does not lead to grid alignment. We also show in Fig. S1 that an architecture represented by 20 overlapping 1DL attractors, each formed by concatenating 10 random cells, aligns grid cells to levels slightly lower but similar to the 1D or 1DL attractors. This architecture can perhaps be considered as simpler to build in biological terms than all the others, but it is still constituted by continuous attractors.

The strength of recurrent collaterals, or more precisely the recurrent to feedforward ratio, is crucial in our model to achieve a negotiated outcome from constraints imposed by the attractor and the inputs. We now show explicit measures of this ratio in Fig. S2, as well as examples showing that an imbalance in this ratio impairs grid alignment. When the ratio is too high or too low, both individual and population gridness are low. Interestingly, grid spacing behaves differently, decreasing monotonically with the relative strength of recurrent connections.

(3) I did not understand what is learned from the local topology analysis. Given that all the grid cells are driven by an input from place cells that spans a 2d manifold, and that the activity in the grid cell network settles on a steady state that depends only on the inputs, isn't it quite obvious that the manifold of activity in the grid cell layer would have, locally, a 2d structure?

The dimensionality of the input is important, although not the only determinant of the topology of the activity. The recurrent collaterals are the other determinant, and their architecture is a crucial feature. For example, as we now show in Figure S2b-d, shuffled recurrent synaptic weights fail to align grid cells. In the 1D condition, if feedforward inputs were absent, the dynamics of the activity would be confined to a ring. The opposite condition is our ‘no attractor’ condition, in which activity in the grid cell layer mimics the topology of inputs, a 2D sheet (and not a torus). It is in the intermediate range, when both feedforward and recurrent inputs are important, that a negotiated solution (a torus) is achieved.

The analyses of local dimensionality and local homology of Figure 3 are crucial steps to demonstrate toroidal topology. According to the theorem of classification of closed surfaces, global homology is not enough to univocally define the topology of a point cloud, and thus this step cannot be skipped. The step is aimed to prove that the point cloud is indeed a closed surface.

(4) The modeling is all done in planar 2d environments, where the feedforward learning mechanism promotes the emergence of a hexagonal pattern in the single neuron tuning curve. This, combined with the fact that all neurons develop spatial patterns with the same spacing and orientation, implies even without any topological analysis that the emerging topology of the population activity is a torus.

We cannot agree with this intuition. In the ‘no attractor’ condition, individual maps have hexagonal symmetry with standardized spacing, but given the lack of alignment the population activity is not a closed surface and thus not a torus. It can rather be described as a 2D sheet embedded in a high dimensional space, a description that also applies to the input space.

While it is rather evident that an ad hoc toroidal architecture folds this 2D population activity into a torus, it is less evident and rather surprising that 1D architectures have the same capability. This is the main novelty in our work.

(5) Moreover, the recent work of Gardner et al. demonstrated much more than the preservation of the topology in the different environments and in sleep: the toroidal tuning curves of individual neurons remained the same in different environments. Previous works, that analyzed pairwise correlations under hippocampal inactivation and various other manipulations, also pointed towards the same conclusion. Thus, the same population activity patterns are expressed in many different conditions. In the present model, the results of Figure 6 suggest that even across distinct rectangular environments, toroidal tuning curves will not be preserved, because there are multiple possible arrangements of the phases on the torus which emerge in different simulations.

We agree with the reviewer in the main point, although the recently found ring activity in the absence of sensory feedback (Gonzalo Cogno et al, 2023) suggests that what is happening in the EC is more nuanced than a pre-wired torus. Solutions in Figure 6 are different ways of folding a 1D strip into a torus, with or without the condition of periodicity in the 1D strip. Whether or not these different solutions would be discernible from one another in a practical setup is not clear to us. For example, global homology, as addressed in the Gardner paper, is the same for all these solutions. Furthermore, while our solutions of up to order 3 are highly discernable, higher order solutions, potentially achievable with other network parameters, would be impossible to discern by eye in representations similar to the ones in Figure 6. In addition, while we chose to keep our model in the simplest possible form as a clear proof of principle, new elements introduced to the model such as head directionality could break the symmetry and lead to the prevalence of one preferred solution for all simulation replicates. We plan to investigate this possibility in the future when attempting to incorporate path-integration capabilities to the model.

(6) In real grid cells, there is a dense and fairly uniform representation of all phases (see the toroidal tuning of grid cells measured by Gardner et al). Here the distribution of phases is not shown, but Figure 7 suggests that phases are non uniformly represented, with significant clustering around a few discrete phases. This, I believe, is also the origin for the difficulty in identifying the toroidal topology based on the transpose of the matrix M: vectors representing the spatial response patterns of individual neurons are localized near the clusters, and there are only a few of them that represent other phases. Therefore, there is no dense coverage of the toroidal manifold that would exist if all phases were represented equally. This is not just a technical issue, however: there appears to be a mismatch between the results of the model and the experimental reality, in terms of the phase coverage.

As mentioned in the results section, Figure 7 is meant for visualization purposes only, and serves more as cautionary tale regarding the imprevisible risks of non-linear dimensionality reduction than as a proof of the organization of activity in the network. Isomap is a non-linear transformation that deforms each of our solutions in a unique way so that, while all have the topology of a torus embedded in a high dimensional space, only a few of them exhibited one of two possible toroidal visualizations in a 3D Isomap reduction. Isomap, as well as all other popular dimensionality reduction techniques, provide no guarantee of topology invariance. A better argument to judge the homogenous distribution of phases is persistent homology, which identifies relatively large holes (compared to the sampling spacing) in the original manifold embedded in a high dimensional space. In our case, persistent homology identified only two holes significantly larger than noise (the two cycles of a torus) and one cavity in all conditions that included attractors. Regarding the specific distribution of phases in different conditions, however, see our reply below.

(7) The manuscript makes several strong claims that incorrectly represent the relation between experimental data and attractor models, on one hand, and the present model on the other hand. For the latter, see the comments above. For the former, I provide a detailed list in the recommendations to the authors, but in short: the paper claims that attractor models induce rigidness in the neural activity which is incompatible with distortions seen in the spatial response patterns of grid cells. However, this claim seems to confuse distortions in the spatial response pattern, which are fully compatible with the attractor model, with distortions in the population activity patterns, which would be incompatible with the attractor model. The attractor model has withstood numerous tests showing that the population activity manifold is rigidly preserved across conditions - a strong prediction (which is not made, as far as I can see, by feedforward models). I am not aware of any data set where distortions of the population activity manifold have been identified, and the preservation has been demonstrated in many examples where the spatial response pattern is disrupted. This is the main point of two papers cited in the present manuscript: by Yoon et al, and Gardner et al.

First of all, we would like to note that our model is a continuous attractor model. Different attractor models have different outcomes, and one of the main conclusions of our manuscript is that attractors can do a wider range of operations than previously thought.

We agree with the reviewer that distortions in spatial activity (which speak against a purely path-integration guided attractor) should not be confused with distortions in the topology of the population activity (which would instead speak against the attractor dynamics itself). We have rephrased these observations in the manuscript. In fact, we believe that the capacity of grid cells to present distorted maps without a distortion of the population activity topology, as shown for example by Gardner and colleagues, could result from a tension between feedforward and recurrent inputs, the potential equilibriums of which our manuscript aims to characterize.

(8) There is also some weakness in the mathematical description of the dynamics. Mathematical equations are formulated in discrete time steps, without a clear interpretation in terms of biophysically relevant time scales. It appears that there are no terms in the dynamics associated with an intrinsic time scale of the neurons or the synapses, and this introduces a difficulty in interpreting synaptic weights as being weak or strong. As mentioned above, the nature of the recurrent dynamics within the grid cell network (whether it exhibits continuous attractor behavior) is not sufficiently clear.

We agree with the reviewer that our model is rather simple, and we value the extent to which this simplicity allows for a deep characterization. All models are simplifications and the best model in any given setup is the one with the minimum amount of complexity necessary to describe the phenomenon under study. We believe that to understand whether or not a 1D continuous attractor architecture can result in a toroidal population activity, a biophysically detailed model, with prohibitive computational costs, would have been unnecessarily complex. This argument does not intend to demerit biophysically detailed models, which are capable of addressing a wider range of questions regarding, for example, the spiking dynamics of grid cells, which cannot be addressed by our simple model.

**Reviewer #3 (Recommendations For The Authors):**
The work points to an interesting scenario for the emergence of toroidal topology, but the interpretation of this idea should be more nuanced. I recommend reconsidering the claims about limitations of the attractor theory, and acknowledging the limitations of the present theory.I don't see the limitations mentioned above as a reason to reject the ideas proposed in this manuscript, for two main reasons: first, additional research might reveal a regime of parameters where some issues can be resolved (e.g. the clustering of phases). In addition, the mechanism described here might act at an early stage in development to set up initial dynamics along a toroidal manifold, while other mechanisms might be responsible for the rigidity of the toroidal manifold in an adult animal. But all this implies that the novelty in the present manuscript is weaker than implied, the ability to explain experimental observations is more limited than implied, and these limitations should be acknowledged and discussed.I recommend reporting on the distribution of grid cell phases and, if indeed clustered, this should be discussed. It will be helpful to explore whether this is the reason for the difficulty in identifying the toroidal topology based on the collection of spatial response patterns (using the transpose of the matrix M).Ideally, a more complete work would also explore in a more systematic and parametric way the influence of the recurrent connectivity's strength on the learning, and whether a toroidal manifold emerges also in non-planar, such as the wagon-wheel environment studied in Gardner et al.

Part of these recommendations have been addressed in the previous points (public review). Regarding the reason why the transpose of M does not fully recapitulate architecture with our conservative classification criteria, we believe that there is no reason why it should in the first place. We view the fact that the transpose of M recapitulates some features of the architecture as a purely phenomenological observation, and we think it is important as a proof that M is not exactly the same for the different conditions. We imagined that if M matrices were exactly the same this could be due to poor spatial sampling by our bins. Knowing that they are intrinsically different is important even if the reason why they have these specific features is not fully clear to us.

Although we do not think that the distribution of phases is related to the absence of a cavity in the transpose of M or to the four clusters found in Isomap projections, it remains an interesting question that we did not explore initially. We are now showing examples of the distribution of phases in Figure S1. We observed that in both 2D and 1D conditions phases are distributed following rather regular patterns. Whether or not these patterns are compatible with experimental observations of phase distribution is to our view debatable, given that so far state-of-the-art techniques have only allowed to simultaneously record a small fraction of the neurons belonging to a given module. This said, we think that it is important to note that ordered phase patterns are an anecdotal outcome of our simulations rather than a necessary outcome of flexible attractors or attractors in general. To prove this point, we simulated a condition with a new architecture represented by the overlap of 20 short 1DL attractors, each recruiting 10 random neurons from the pool of 100 available ones.

The rest of the parameters of the simulations were identical to those in the other conditions.

By definition, the topology of this architecture has Betti numbers [20,0,0]. We show in Figure S1 that this architecture aligns grid cells, with individual and population gridness reaching slightly lower levels compared to the 1D condition. However, the distribution of phases of these grid cells has no discernible pattern. This result is an arbitrary example that serves as a proof-of-principle to show that flexible attractors can align grid cells without exhibiting ordered phases, not a full characterization of the outcome of this type of architecture, which we leave for future work. For the rest of our work, we stick to the simplest versions of 1D architectures, which allow for a more in-depth characterization.

The wagon-wheel is an interesting case in which maps loose hexagonal symmetry although the population activity lies in a torus, perhaps evidencing the tension between feedforward and recurrent inputs and suggesting that grid cell response does not obey the single master of path integration. If we modeled it with a 1D attractor, we believe the outcome would strongly depend on virtual rat trajectory. If the trajectory was strictly linear, the population activity would be locally one-dimensional and potentially represented by a ring. Instead, if the trajectory allowed for turns, i.e. a 2D trajectory within a corridor-like maze, the population activity would be toroidal as in our open field simulations, while maps would not have perfect hexagonal symmetry, mimicking experimental results.

More minor comments:Recurrent dynamics are modeled as if there is no intrinsic synaptic or membrane time constant. This may be acceptable for addressing the goals of this paper, but it is a bit unusual and it will be helpful to explain and justify this choice.

As mentioned above, we believe that the best model in a given setup is the one with the lowest number of complexities that can still address the phenomenon under study. One does not use general relativity to build a bridge, although it provides a ‘more accurate’ description of the physics involved. All models are simplifications, and the more complex a model, the more it has to be taken as a black box.

The Introduction mentions that in most models interaction between co-modular neurons occurs through direct excitatory communication, but in quite a few models the interaction is inhibitory. The crucial feature is that the interaction is strongly inhibitory between neurons that differ in their tuning, and either less inhibitory or excitatory between neurons with similar phases.

We agree that directed inhibition has been shown to be as efficient as directed excitation, and we have modified the introduction to reflect this.

The Discussion claims that the present work is the first one in which the topology of the recurrent architecture differs from the topology of the emergent state space. However, early works on attractor models of grid cells showed how neural connectivity which is arranged on a 2d plane, without any periodic boundary conditions, leads to a state space that exhibits the toroidal topology. Therefore, this claim should be revised.

We agree, although the 2D sheet in this case acts as a piece of the torus, and locally the input space and architecture are identical objects. It could be argued that architectures that represent a 2D local slice of the torus, the whole torus, or several cycles around the torus form a continuous family parametrized by the extension of recurrent connections, and as a consequence it is not surprising that these works have not made claims about the incongruence between architecture and representation topologies. The 2D sheet connectivity is still constructed ad hoc to organize activity in a 2D bump, and there is no negotiation between disparate constraints because locally the constraints imposed by input and architecture are the same. We believe this situation is conceptually different from our flexible 1D attractors. We have adapted our claim to include this technical nuance.

Why are neural responses in the perimeter of the environment excluded from the topological analysis? The whole point of the toroidal manifold analysis on real experimental data is that the toroidal manifold is preserved regardless of the animal's location and behavioral condition.

We agree, although experimental data needs to go through extensive pre-processing such as dimensionality reduction before showing a toroidal topology. Such manipulations might smooth away the specific effects of boundaries on maps, together with other sources of noise. In our case, the original reason to downsample the dataset is related to the explosion in computational time that we experience with the ripser package when using more than ~1000 data points. For a proof-of-principle characterization we were much more interested in what happened in the center of the arena, where a 1D attractor could fold itself to confine population activity into a torus. The area we chose was sufficiently large to contain the whole torus. Borders do affect the way the attractor folds (they also affect grid maps in real rats). We feel that these imperfections could be interesting to study in relation to the parameters controlling how our virtual rat behaves at the borders, but not at this proof-of-principle stage.

The periodic activity observed in Ref. 29 could in principle provide the basis for the ring arrangement of neurons. However, it is not yet clear whether grid cells participate in this periodic activity.

We agree. So far it seems that entorhinal cells in general participate in the ring, which would imply that all kinds of cells are involved. However, it could well be that only some functional types participate in the ring and grid cells specifically do not, as future experiments will tell.